# Root uptake under mismatched distributions of water and nutrients in the root zone.

Jing Yan[1], Nathaniel A. Bogie[2], and Teamrat A. Ghezzehei[1]

[1]Life and Environmental Sciences Department, University of California, Merced, CA 95343, USA
[2]Geology Department, San Jose State University, San Jose, CA 95192, USA

**Correspondence:** Teamrat A. Ghezzehei (taghezzehei@ucmerced.edu)

**Abstract.** Most plants derive their water and nutrient needs from soils, where the resources are often scarce, patchy, and ephemeral. It is not uncommon for plant roots to encounter mismatched patches of water-rich and nutrient-rich regions in natural environments. Such an uneven distribution of resources necessitates plants to rely on strategies to explore and acquire nutrients from relatively dry patches. We conducted a laboratory study that elucidates the biophysical mechanisms that enable this adaptation. The roots of tomato (*Solanum lycopersicum* ) seedlings were laterally split and grown in two adjacent, hydraulically-disconnected pots, which permitted precise control of water and nutrient applications to each compartment. We observed that physical separation of water-rich and nutrient-rich compartments (one received 90% water + 0% nutrients and the other received 10% water + 100% nutrients) does not significantly stunt plant growth and productivity compared to two control treatments (control 1: 90% water + 100% nutrients versus 10% water + 0% nutrients; and control 2: 50% water + 50% nutrients in each compartment). Specifically, we showed that soil dryness does not reduce nutrient uptake, vegetative growth, flowering, and fruiting compared to control treatments. We identified localized root proliferation in nutrient-rich dry soil patches as a critical strategy that enabled nutrient capture. We observed nocturnal rewetting of the nutrient-rich but dry soil zone (10% water + 100% nutrients) but not in the nutrient-free and dry zone of the control experiment (90% water + 100% nutrients). We interpreted the rewetting as the transfer of water from the wet to dry zones through roots, a process commonly known as hydraulic redistribution (HR). The occurrence of HR likely prevents the nutrient-rich soil from drying to permanent wilting and subsequent decline of root functions. Sustaining rhizosphere wetness is also likely to increase nutrient mobility and uptake. Lack of HR in the absence of nutrients suggests that HR is not entirely passive, water-potential gradient driven flow. The density and size of root-hairs appeared to be higher (qualitative observation) in the nutrient-rich and dry compartments than the nutrient-free and dry compartments. We also observed organic coating on sand grains in the rhizosphere of the nutrient-rich and dry compartments. The observations are consistent with prior observations that root hairs and rhizodeposition aid rhizosphere wetting. These findings were synthesized in a conceptual model that explains how plants of dry regions may be adapted to mismatched resources. This study also suggests that separating the bulk of applied nutrients from the frequently irrigated soil region can increase nutrient use efficiency and curtail water pollution from intensive agricultural systems.

# 1 Introduction

Root response to either water or nutrient deficiency signals is a persistent question at the intersection of plant biology and soil science (Robbins and Dinneny, 2015; Hodge, 2004; Robinson et al., 1999). In water-limited areas, rooting depth generally coincides with infiltration depth (Fan et al., 2017). Locally, roots also respond by increasing the water retention capability of their immediate surroundings (the rhizosphere) by releasing a cocktail of organic compounds (rhizodeposits) that sorb water and promote soil aggregation (Carminati et al., 2010, 2011; Moradi et al., 2011; Albalasmeh and Ghezzehei, 2014; Ghezzehei and Albalasmeh, 2015). Similarly, roots employ diverse strategies of nutrient foraging in response to local soil nutrient deficiencies and macroscopic heterogeneities. Roots can enlarge the surface area for nutrient sorption and acquisition by increasing root branching, clustering, and growing dense root hairs (Lambers et al., 2011; Bates and Lynch, 2001). Legumes associate with N-fixing bacteria and mycorrhizae that support N fixation and acquisition (Linderman, 1991), while non-legume plants forage by growing their roots towards their N-fixing legume neighbors (Weidlich et al., 2018). Root exudation (release of low-molecular-weight rhizodeposits (Oburger and Jones, 2018)) can increase nutrient availability and accessibility by freeing tightly-bound nutrients (e.g., McKay Fletcher et al., 2020) and priming microbial mineralization of nutrients (e.g., Keiluweit et al., 2015).

However, the adaptation of roots to mismatched distributions of water and nutrients has not received as much attention. Spatial and temporal mismatch of water and nutrient availability within the soil profile is a frequent occurrence confronting plants in regions with pronounced wetting-drying cycles (Bengough et al., 2011). A recent review (Fan et al., 2017) of global rooting depth data in well-drained upland environments and drought-prone regions revealed that soil moisture distribution within the soil profile is the primary determinant of root architecture. Plants that grow in such areas meet the bulk of their transpiration demand with subsurface soil moisture storage because the shallow soil layers tend to dry up quickly by evaporation and drainage. Coarse-textured soils that dominate most arid and semi-arid soil soils (Rodell et al., 2004) experience particularly pronounced surface drying. In contrast, organic matter and plant-available essential macro- and micro-nutrients, including N, P, K, Zn, Mn, are preferentially concentrated in the shallow soil horizons (Li et al., 2013; Jobbágy and Jackson, 2001; Franzluebbers and Hons, 1996; Apostolakis and Douka, 1970). This mismatched distribution of water and nutrients necessitates nutrient uptake from relatively dry, but nutrient-rich, soil patches (Nambiar, 1976; Rose et al., 2008; Wang et al., 2009). Moreover, the release of nutrients bound in organic matter of the shallow soil layers requires mineralization to occur under sub-optimal moisture conditions (Stanford and Epstein, 1974). The adaptation of many plants to these environments suggests the existence of nutrient and water acquisition strategies that allow root architecture and functions to respond to mismatched spatial and temporal nutrient and water distributions. In addition to natural systems, such adaptation is likely to play a critical role in dry-land farming and rangelands.

The conditions responsible for mismatched resource distributions are also favorable to the transport of water from the wet subsurface layers to dry shallow layers via the root system, commonly referred to as hydraulic-lift or hydraulic-redistribution (HR) (Caldwell and Richards, 1989; Bogie et al., 2018). Studies have found that water released by HR can elevate ammonification, N mineralization, and plant inflorescence N uptake (Cardon et al., 2013) and enhance the overall nutrient mobility

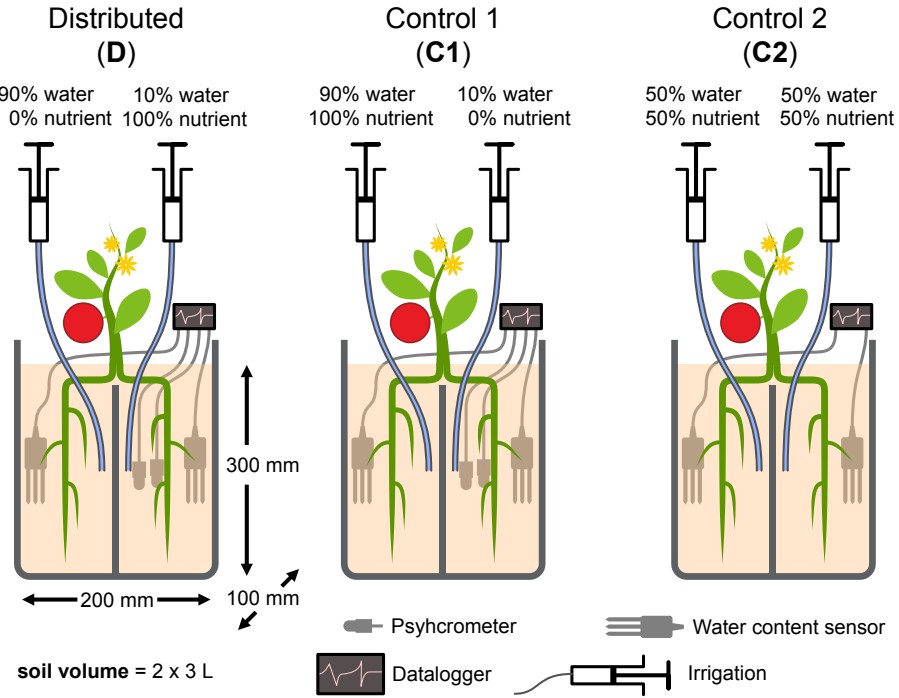

**Figure 1.** Schematic illustration of the experimental design. Each pot consists of two isolated compartments that are fused together by glue, that were supplied water and nutrients via buried Nylon tubing. The relative amounts of applied water and nutrients are shown. Roots of seedlings were roughly divided half and half during transplantation. The experiment consisted of one treatment in which the bulk quantity of water and nutrients were distributed separately (treatment **D**) and two control treatments in which nutrients were applied with most of the water. In Control 1 (**C1**) water was applied non-uniformly as in **D**, where in Control 2 (**C2**), water and nutrients were applied uniformly to both compartments. Placement of sensors and water and nutrient delivery tubes are illustrated. Diagram is not to scale.

in dry soil patches (Matimati et al., 2014). The objective of this study was to test the hypothesis that HR is a key biophysical response that allows plants to thrive when resource availabilities are spatially mismatched. Specifically, the laboratory experiments were designed to answer the following questions: Does the mismatched distribution of water and nutrients within a soil profile adversely affect plant performance? To what extent are roots able to acquire nutrients from dry soil patches provided that water is available elsewhere? What is the role of HR in nutrient uptake from dry patches?

## 2 Methods

### 2.1 Experimental setup

Tomato plants (*Solanum lycopersicum*) were grown in custom pots ($W \times D \times H = 200 \times 100 \times 300$ mm) that were laterally split into two equal (3 L each) compartments (see Figure 1). The pots were filled with 8 kg silica sand of approximate median

**Table 1.** Total quantity of water and N applied to each compartment of the three treatments. Note that the nutrient applied nutrient solution includes other macro and macro nutrients. The composition of the nutrient solution is provided in Table A2.

| Treatment | Code | Applied Water (mm) | | Applied N (mgN) | |
|---|---|---|---|---|---|
| | | Wet | Dry | Wet | Dry |
| Distributed | D | 588 | 77 | 0 | 120 |
| Control 1 | C1 | 580 | 73 | 120 | 0 |
| | | Left | Right | Left | Right |
| Control 2 | C2 | 338 | 338 | 60 | 60 |

particle size of $600\,\mu m$ (Laguna Clay Co., City of Industry, CA), and packed to mean bulk density of $1.4\,g\,cm^{-3}$. The sand was free of nutrients and organic matter to ensure that all the nutrient supply was accounted for. Nylon tubings for water and nutrient solution injection were installed in each compartment at $140\,mm$ below the surface. Dielectric water content sensors (5TE of Meter, Pullman, WA) were placed at the center of each compartment (center of sensors was at $140\,mm$ below the surface) to capture the bulk-scale soil moisture dynamics. At the same soil depth, the dry compartments of treatment (see below for treatment descriptions) were outfitted with pairs of thermocouple psychrometric water potential sensors (Psypro of Wescor Inc. Logan, UT) to measure the localized soil water potential with a high degree of sensitivity (Brown and Bartos, 1982; Andraski and Scanlon, 2002; Whalley et al., 2013). The combination of the two sensor types allows quantification of water dynamics with high degree of fidelity from the wet to dry moisture range. The dielectric sensors were programmed to log data every $15\,min$ while the psychrometers were programmed to log data every $2\,h$. The experiment was conducted indoors (in a dark room) under artificial fluorescent lighting ($6,500\,K$ spectrum and $10,000\,lm$ intensity) that was programmed to be on for $12\,hr$ and off for $12\,hr$.

## 2.2 Experimental treatments

The experimental design consisted of one primary treatment and two control treatments, that were replicated three times each. While the total amounts of applied resources to the three treatments were identical, the treatments differed in the distributions of water and nutrients between the two compartments. The relative amounts of water and nutrients supplied to each compartment along with the treatment names and codes are depicted in Figure 1. In the primary treatment, the distribution of water and nutrients between the two compartments was mismatched (labeled as Distributed or **D**). One compartment (*wet*) received $\approx 90\%$ of the irrigation water and $0\%$ of the nutrients, while the second compartment (*dry*) received $100\%$ of the nutrient supply delivered along with the remaining $10\%$ water. The first control treatment (labeled as Control 1 or **C1**) consisted of identical distribution of water as treatment **D**. But the nutrients were added to the wet compartment along with the $\approx 90\%$ irrigation water, while the dry compartment remained nutrient-free. In the second control experiment (labeled as Control 2 or **C2**), both compartments (*left and right*) received equal amounts of water and nutrients. The contrast between treatment **D** and **C1** was

intended to reveal plant response and adaptation mechanisms to the mismatched distribution of resources. Whereas the contrast between mismatched distribution in treatment **D** and the ideal uniform resource availability in treatment **C2** was intended to identify possible adverse effects of the former. Deionized water was used for irrigation. The nutrient solution was prepared by diluting commercial hydroponic nutrient solution (General Hydroponics, Santa Rosa, CA), that consisted of $NH_4NO_3$, $Ca(NO_3)_2$, $Mg(NO_3)_2$, $MgSO_4$, $KH_2PO_4$, $KNO_3$, $K_2SO_4$, and $Na_2MoO_4$. The relative mass-based elemental composition of the nutrient solution (General Hydroponics, Santa Rosa, CA) normalized against the total N content is provided in Table A2. The total amounts of water and nutrients supplied to each compartment are reported in Table 1, while the detailed record is provided in SI data sets.

The tomato seedlings used for the study were germinated in potting mix and grown for about 3 weeks until they reached 50-100 mm in height. The healthiest seedlings were then removed from the pots, and the roots were thoroughly washed to remove any residual nutrient from the potting mix. The roots of individual seedlings were then physically separated into two roughly equal parts that were placed in the separate compartments, which were half-filled with soil. Afterward, the remainder sand was carefully poured around and over the roots. We did not differentiate the taproots from the fibrous roots during the root splitting. A photographic depiction of the key steps of the experiment, including root splitting, sensor installation, and irrigation outfitting, is provided in SI (Figure S1-S4). After the transplantation, the pots were irrigated with 560 ml of deionized water (equivalent to 0.2 v/v). The plants were allowed to adjust to the new environment for around 2 weeks with no additional irrigation or fertilization. Subsequently, the prescribed application of water and nutrient solution commenced on the day 18 after transplantation. The experiment lasted for 140 days after transplantation.

## 2.3 Plant and soil characterization

The plants were harvested on days 138 to 140 after transplantation and multiple indicators of plant performance were measured. Shoot dry mass determined separately for each branch of every plant. The number of flowers, number of fruits, and fruit dry mass were determined for each plant. Leaf greenness within individual plant canopy was evaluated in terms of normalized difference vegetation index (NDVI) captured by hyperspectral analyses of leaf samples (ASD Spectroradiometer, Malvern Panalytical, Cambridge, UK)

$$\text{NDVI} = \frac{R_{800} - R_{670}}{R_{800} + R_{670}}$$

where $R_{670}$ and $R_{800}$ are reflectance intensity at 670 and 800 nm and represent red and near-infrared lights, respectively.

Shoot and fruit N content across canopy were determined using a high-temperature combustion elemental analyzer (Thermo Fisher Scientific, Waltham, MA). The total plant N uptake by the aboveground biomass was calculated by integrating dry-mass-weighted N content for each plant. Total N uptake was corrected by subtracting the initial plant total N mass of representative seedling. The N use efficiency (NUE) of the aboveground biomass was calculated as the ratio of net plant N uptake to total N addition during the experiment. The N content of the sand medium before and after the experiment was below the detection limit.

After harvest, the soil compartments were allowed to air-dry until the water content reached 3.5% to 4% by volume. A uniform soil drying condition was established by subjecting the pots to constant airflow inside a fume hood. Then the soil from

one replicate of each treatment was carefully scooped out at $20\,\mathrm{mm}$ depth intervals. The coarse root pieces in each interval were kept in place by cutting. Roots were removed by gentle sieving ($4\,\mathrm{mm}$ mesh) and subsequent and manual picking. The dry root mass in each compartment is reported. In separate replicates, the sand-coated roots (rhizosheaths) were preserved by removing roots with minimum agitation and were used for microscopic analysis. Confocal images were obtained using a Zeiss LSM $880$ Airyscan confocal microscope and EC Plan-Neofluar 10x/0.30NA objective lens (Carl Zeiss Microscopy LLC, White Plains, NY). We used $405\,\mathrm{nm}$ and $488\,\mathrm{nm}$ lasers to excite and identify autofluorescent organic compounds from the non-fluorescent soil matrix. T-PMT detector was used to acquire transmitted light images. Detailed morphology of the roots and root-hairs was acquired using Scanning Electron Microscopy (SEM) (Zeiss Gemini SEM 500, Carl Zeiss Microscopy LLC, White Plains, NY). SEM images were acquired at $3\,\mathrm{kV}$ after coating the samples with gold (E5000 Sputter Coater, Quorum Technologies Ltd, East Sussex, UK). A homogenized gold coating was used to provide a conductive layer of metal that enhances image quality by preventing charging and damage of biological tissues (Kim et al., 2010; Golding et al., 2016). Image analysis and processing was done using ImageJ (Schneider et al., 2012).

## 2.4   Soil hydraulic properties and fluxes

The water retention curve of silica sand was determined by water potentiometer (WP4C, Meter Group, Pullman, WA). To account for the osmotic effect of the nutrients on water potential, we used a nutrient solution of $520\,\mathrm{mgN/L}$ that was consistent with the nutrient solution added to the dry compartment of treatment **D**). The resulting water retention curves were fitted with Brooks Corey model (Brooks and Corey, 1966, 1964)

$$\theta/\theta_S = (\psi/\psi_0)^\lambda$$

where $\psi_0$ is air-entry water potential, $\theta_S$ is the saturated water content, and the residual water content was assumed to be zero. The fitted water retention curve was used to convert psychrometric water-potential readings to equivalent volumetric water content. The water potential range necessary for HR calculations is $\psi \leq -100\,\mathrm{kPa}$. Because the water-potential derived moisture dynamics is more sensitive to small changes than the dielectric sensors, it was used for estimation HR.

The magnitude of HR flux was defined as the water flux released from the root surface to the soil ($\mathrm{mm/day}$). The gain in volumetric water content was calculated by subtracting minimum daily root-zone water content from the subsequent daily maximum values (Meinzer et al., 2004). The volumetric water content was then scaled to an equivalent soil moisture depth by multiplying it by the thickness of the layer where roots were concentrated. Only the data after the plants were well established and the effect of the initial moisture has disappeared were used for this estimation. The mean soil water potential corresponding to the observed HR was calculated as the arithmetic average of the data recorded between noon of the preceding day and the day HR occurred.

## 2.5   Statistical and data analysis

Plant physiological indicators were compared across treatments using a Welch's analysis of variance (ANOVA) to avoid interference from heteroscedasticity of those indicators (Welch, 1947). Posthoc multiple comparisons were performed using the

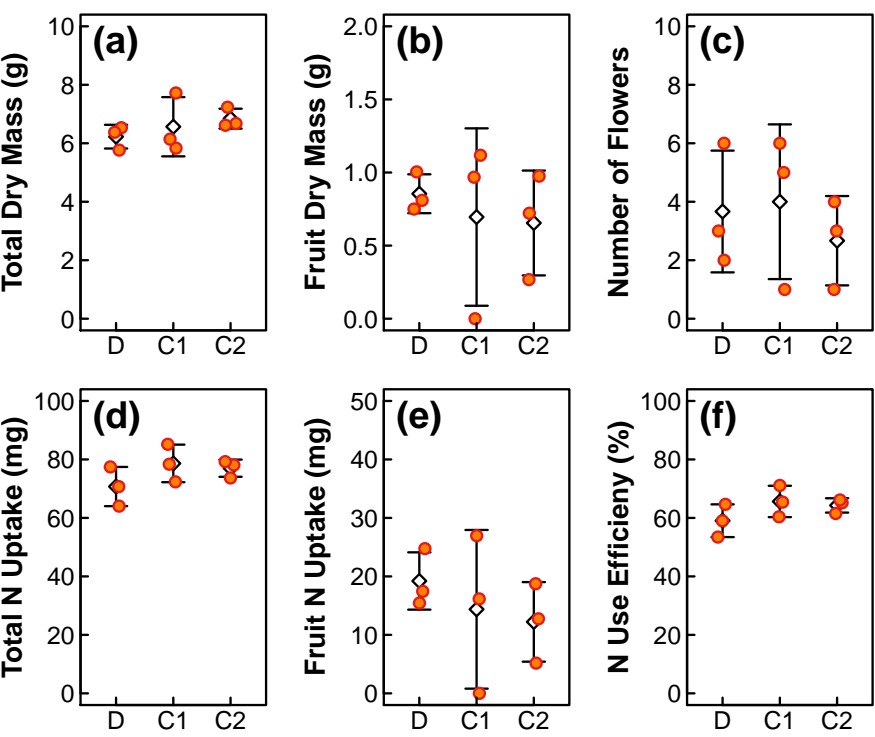

**Figure 2.** Comparison of plant physiological indicators (a) total dry biomass, (b) fruit dry mass, (c) number of flowers, (d) total N uptake, (e) N uptake in Fruits, and (f) N use efficiency in treatment **D**, **C1**, and **C2**. The orange dots represent values of individual replicates. The white diamonds and whiskers represent the mean and standard deviation within each treatment. Distribution of N content along the canopy length is shown in Figure 3. One of the replicates in treatment **C1** did not produce fruits, resulting in larger deviations in fruit dry mass and N update in treatment **C1**.

Games-Howell test (Games and Howell, 1976). Similarly, NDVI and N content at the whole-plant scale were compared across treatments using Welch's ANOVA test. Mature leaf samples, i.e. the 3rd to 6th leaves from the growing tip (equivalent to the normalized plant height of 0.8 to 0.9), were further selected to assess the greenness of mature leaf as suggested by Kalra (1998) (reported as Leaf NDVI $0.8 - 0.9$ in Table A1).

## 3 Results

### 3.1 Above-ground plant characteristics

The means of total aboveground biomass, fruit mass, number of flowers, N uptake by total biomass and fruits, and NUE within each treatment are reported in Figure 2. Error bars indicate standard deviations of three replicates. The means, standard deviations, and $p$-value of Welch's ANOVA test of these variables and other whole-plant-scale indicators of performance are

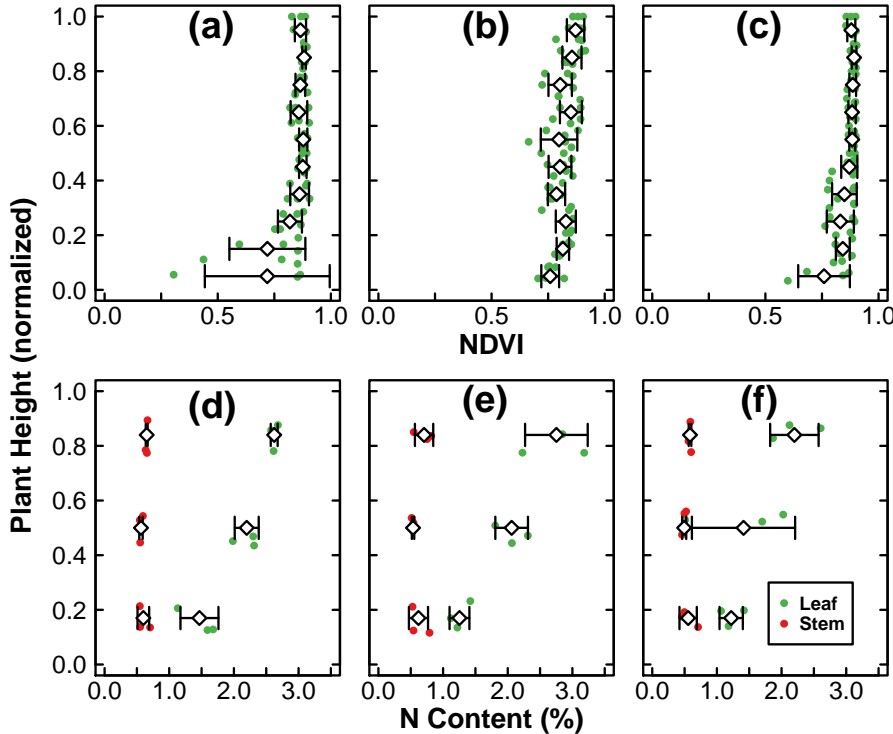

**Figure 3.** Leaf NDVI as a function of normalized plant height at the end of the experiments in treatment **D** (a), **C1** (b), and **C2** (c); N content (%) of stem and leaf samples across canopy at the end of the experiments in treatment **D** (d), **C1** (e), and **C2** (f). The green dots represent leaf samples, while the red dots represent stem samples. The dots include three replicates within each treatment. The diamonds and whiskers represent the mean and standard deviation of replicates at the normalized plant height. Note: mean and standard deviation of leaf NDVI was calculated within an incremental height of 0.1; N content (%) of stem and leaf samples were separated into three portions across the canopy and thus reported as the normalized height of 0.17, 0.5 and 0.84.

reported in Table A1. None of the indicators had statistically significant differences between treatment means determined by Welch's one-way ANOVA ($p > 0.05$). Distributions of tissue N content and leaf greenness across plant canopy are reported in Figure 3. The vertical axis represents the normalized plant heights. In Figure 3a, the distance of the leaf stem from the soil surface is rounded to the nearest tenths; then, the NDVI of leaves of the same height are grouped to calculate the mean and standard deviation. The N content of leaves and stems were similarly pooled into three height-groups. Within each height

group, there were no statistically significant differences between treatment means of N content determined by Welch's one-way ANOVA ($p > 0.05$). There was a statistically significant difference in whole-plant-NDVI between treatment means determined by Welch's one-way ANOVA ($p < 0.001$). Further, pair-wise posthoc comparison showed that the whole-plant-NDVI in treatment **D** was not significantly different from the other controls, but there was a significant difference between the means of the two controls ($p < 0.001$). Additional details are reported in Table A1. N concentration and NDVI varied considerably with

plant height within each treatment. Generally, the younger leaves at the top of the canopy have higher N content and NDVI

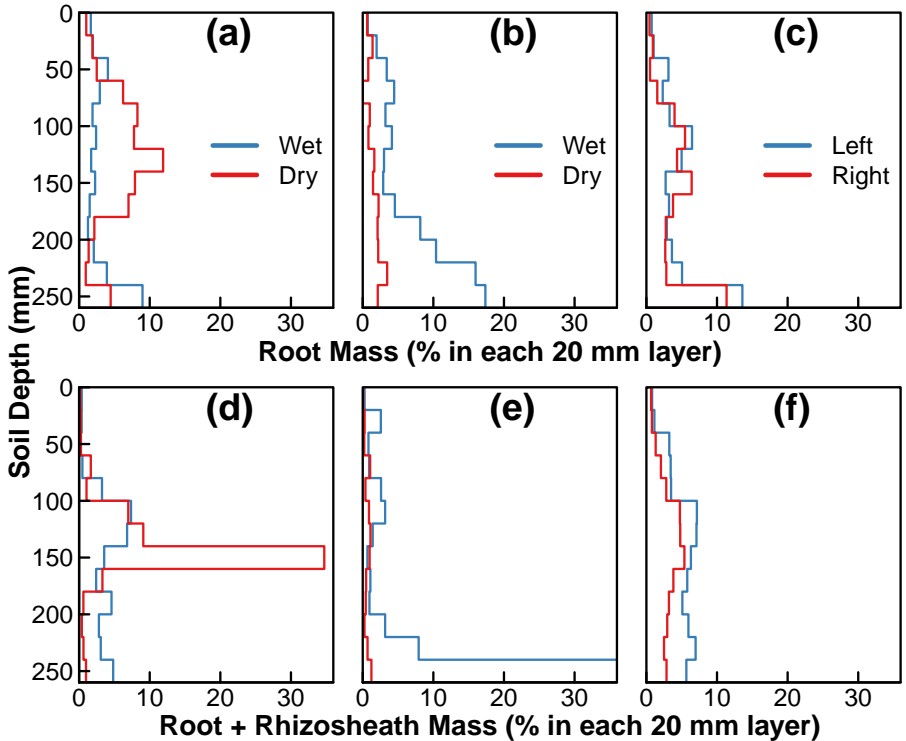

**Figure 4.** Incremental root mass distribution and distribution of total mass of root and rhizosheath along the soil profile in treatment **D** (a, d), **C1** (b, e) and **C2** (c, f). The root mass and total mass of root and rhizosheath within each interval was extracted and normalized to the total root or root and rhizosheath mass from the two isolated compartments. Each step in the plot represents the normalized mass distribution within the 20 mm soil depth. Note: "Wet" and "Dry" compartments (compartments with 90% versus 10% water, respectively in Figure 1) were defined operationally to distinguish water supply for treatment **D** and **C1** mainly; in treatment **C2**, the water was supplied uniformly in the disconnected compartments, and they are labeled as 'Left' and "Right" compartments. Detailed schemes of water and nutrient supply were provided in Figure 1.

than the oldest leaves at the base (Figure 3). Such nutrient translocation during plant growth is typical. Therefore, we did not perform a statistical test of height-based differences.

### 3.2    Root distribution and rhizosphere characteristics

Root mass distributions in the two compartments of the three treatments are shown in Figure 4. The values represent the root mass in each 20 mm layer as a percent of the total root mass of the whole plant. The top row is from replicates where all the sand was removed from the rhizospheres. Whereas the bottom row is from replicates in which the sand particles that tightly adhered to the roots (rhizosheath) were kept intact. For treatments **D** and **C1**, the wet and dry labels refer to the compartments that received $\approx 90\%$ and $\approx 10\%$ irrigation water, respectively. Note that there were no such differences in applied irrigation water between the two compartments of treatment **C2** (see Figure 1). The roots in the nutrient-free wet compartment of treatment **D**

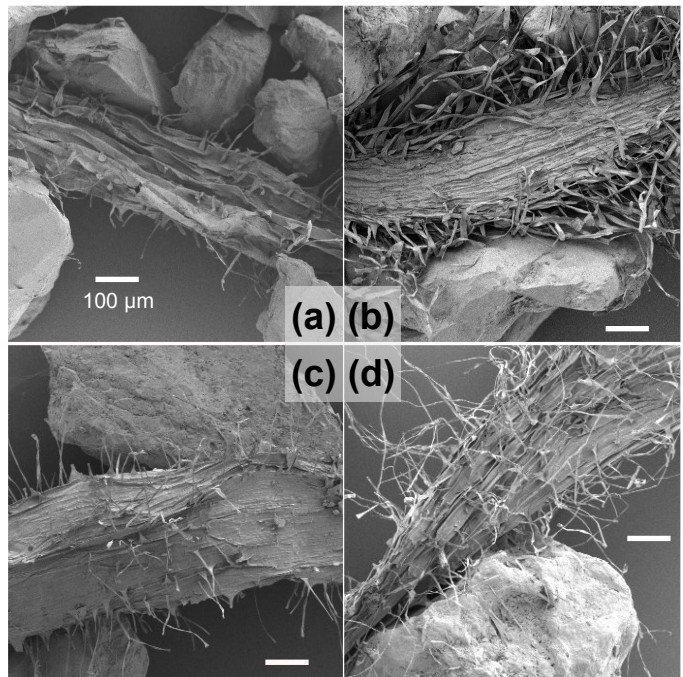

**Figure 5.** SEM images of representative rhizosheaths collected from the "Wet" and "Dry" compartments of treatment **D** (a and b, respectively) and **C1** (c and d, respectively). All the SEM images have identical magnification (all four subfigures used a $100\ \mu m$ scale bar) that permits visual qualitative comparison.

(marked as "wet" in Figure 4 (a)) were distributed mostly uniformly throughout the depth profile, with a slight increase near the bottom. In contrast, the roots grown in the nutrient-rich "dry" compartment were mainly concentrated in the mid-section, coinciding with the depth at which nutrient solution was supplied using a subsurface injector (Figure 1). Overall, $60\%$ of the total root mass was observed in the nutrient-rich dry compartment of treatment **D**, where only $\approx 10\%$ of the irrigation water was available ("dry" compartment in Figure 4a). In contrast, the root mass distribution in the nutrient-rich wet compartment of the first control treatment **C1** increased with depth, with a notable accumulation of root biomass at the base ("wet" compartment in Figure 4b). However, the root mass in the nutrient-free dry compartment was stunted and accounted for only $20\%$ of the total root mass ("dry" compartment in Figure 4b). There was no considerable difference in root mass distribution between the two compartments of the second control treatment **C2**, where water and nutrient were supplied equally to both compartments. There was a slight accumulation of roots at the base of both compartments.

The replicates with rhizosheaths exhibited similar general patterns as the roots. But two regions showed more pronounced root and rhizosheath accumulation than roots alone. First, the mid-section ($140$ mm to $160$ mm) of the nutrient-rich dry compartment of treatment **D** (Figure 4d) contained one-third of the roots and rhizosheaths of the entire plant. Visually, more roots appeared to be covered by sand in this layer than any other in the entire compartment. Fewer roots exhibited rhizosheaths

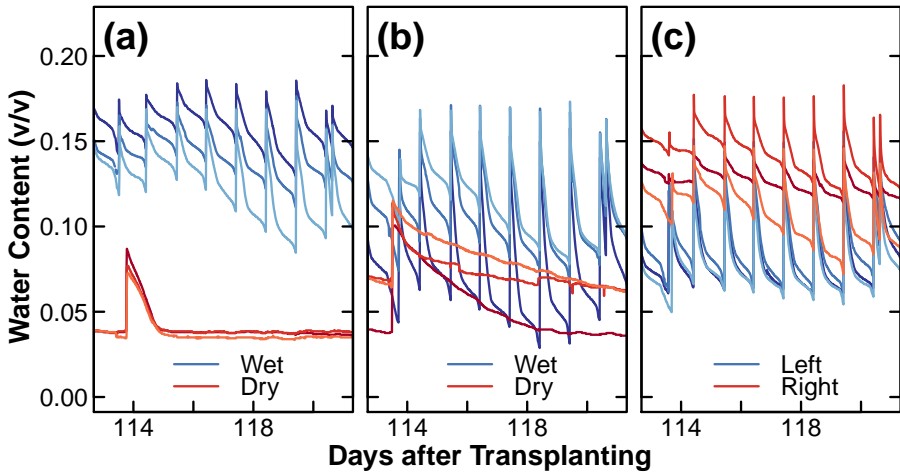

**Figure 6.** Changes in dielectric soil volumetric water content (v/v) during days of 113 to 121 after transplantation in "Wet" and "Dry" compartments of treatment **D**, **C1** and in "Left" and "Right" compartments of treatment **C2** (a, b, c). The different shades of red and blue in these figures are used to distinguish between replicates. Note that the "Wet" compartments were irrigated daily, while the "Dry" compartments were irrigated once a week for the majority of the experiments (days of 40 to 140 after transplantation). The results plotted represent a typical cycle of soil water content changes. The long-term results of dielectric soil volumetric water content can be found in the SI.

in the nutrient-free dry compartment of treatment **C1** (Figure 4e). Therefore, both the root density and intensity of rhizosheath
formation appear to be correlated to nutrient availability in the dry soil.

Similarly, the bottom layer of the wet compartment in treatment **C1** accounted for two-thirds of the roots and rhizosheaths of the entire plant. The roots at this depth visually exhibited by far the highest amount of rhizosheath than any other layer. This is partly due to the ponding of water at the base of this compartment and possible differences in degree of drying.

The roots in all the treatments exhibited the formation of rhizosheath. There were no differences in the visual appearance of
rhizosheaths collected from the wet and dry compartments in the three treatments. However, notable differences were observed in the microscopy images. Representative SEM images of the rhizosheaths of treatments **D** and **C1** are shown in Figure 5. Overall, the dry compartments (Figure 5 b and d) exhibited denser and longer root-hairs than their wet counterparts (Figure 5 a and c) within the same treatments. There was little noticeable difference between the two wet compartments (Figure 5 a and c). The nutrient-rich dry compartment (Figure 5b) exhibited visually thicker and more dense root hairs compared to the nutrient-
free dry compartment (Figure 5d). Confocal microscopy images of the sand grains in the rhizosheath of the nutrient-rich dry compartment of treatment **D** showed extensive amorphous organic coating that appeared to be distinct from the roots and root hairs (Figure S5).

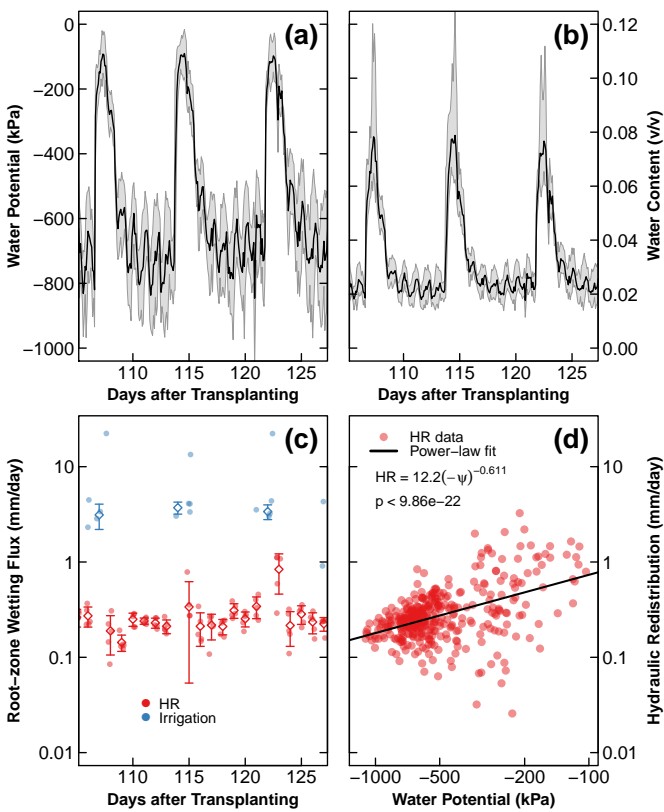

**Figure 7.** Changes in soil water potential (a), water content converted from soil water potential (b), and root-zone wetting flux (c) from HR and irrigation as a function of time in "Dry" compartment of treatment **D** during days of 113 to 121 after transplantation; HR outflow magnitude as a function of water potential ($\psi$): HR described by a power-law model is shown in solid line (d). In (a) and (b), solid black lines and grey shade represented the average and the standard deviation of soil water potential and converted water content from five sensors distributed in three replicate compartments. Similarly, in (c), solid dots represent the calculated water flux from five sensors, and the diamonds and whiskers show the average and standard deviation of the water flux. In (d), water flux from HR during the whole experiment was used. The long-term results of soil water potential and converted water content were provided in the supplemental materials.

### 3.3 Water dynamics

A representative water content dynamics measured by the dielectric sensors during a typical week is shown in Figure 6. This pattern occurred consistently throughout the experiment. The complete dataset is shown in Figure S6). The different shades of blue and red represent the replicates of the wet and dry compartments. Recall that there was no distinction in wetness in treatment **C2**. The observed differences between replicates are likely due to variations in the proximity of sensors to roots and the irrigation tubing or random differences in soil packing and plant growth patterns. Overall, the wet compartments irrigated on a daily cycle remained at a higher moisture level most of the time. The most striking difference was observed between the dry compartments of treatment **D** and **C1**. Specifically, in the presence of nutrients, the water content in the dry compartment of treatment **D** was depleted within one day after each application of nutrient solution (Figure 6a). Whereas the water content of the nutrient-free dry compartment of treatment **C1** declined slowly over a week (Figure 6b). This difference in water uptake rate is consistent with the differences in root mass distribution between the dry compartments shown in Figure 4a and 4b, respectively.

Psychrometers were installed only in the dry compartments of the treatment **D** and **C1**. The sensors in treatment **C1** did not register meaningful data because the soil never dried to within the measurement range of the psychrometers ($\psi < -50 kPa$). Also, one of the three pairs of psychrometers installed in treatment **D** failed. A representative three-week range of data from the functioning five sensors is shown in Figure 7a. The full data set including the unphysical readings of the relatively wet soils is provided in the SI (Figure S7). The soil water potential increased with each nutrient solution application and dried to the pre-irrigation level within one day, consistent with the water content data shown in Figure 6a. One day after applying the nutrient solution, the water potential exhibited a diurnal fluctuation with a daytime decline and a nighttime increase. But the average water potential remained stable during this period. The variation remained between $-100$ and $-1000$ kPa and did not dry to the level of permanent wilting ($\psi - 1500$ kPa).

The water potential data was converted to volumetric water content using a water retention curve of the same soil (Figure A1). The resulting water content dynamics are shown in Figure 7b (see Figure S7 for the full range). Notice that this water content dynamics reveal more detailed diurnal fluctuation than the dielectric sensors with limited sensitivity to small daily changes between $0.01$ and $0.12$ v/v. The nocturnal water release increased the root zone water content by $\approx 0.1$ v/v, and a similar amount was taken up by the roots during day time.

The gain in volumetric water content was multiplied by $50$ mm, which represents the approximate depth of root concentration in the nutrient-rich dry compartment (Figure 4a). The gain water content during nutrient application (irrigation) was also calculated similarly but marked differently. The magnitudes of the root-zone wetting by HR and irrigation fluxes, which were recorded by each of the five working sensors are reported in Figure 7c. HR flux remained consistent for most of the study duration, with slight increases observed in the first day after each irrigation event. The HR flux was one order of magnitude lower than the intermittent irrigation ($0.1 - 1.0$ versus $1.0 - 10$ mm/day). The relationship between the absolute magnitude of HR and the daily mean water potential is shown in Figure 7d. A statistically significant positive trend ($p < 9.8 \times 10^{-22}$) described by a power law was observed.

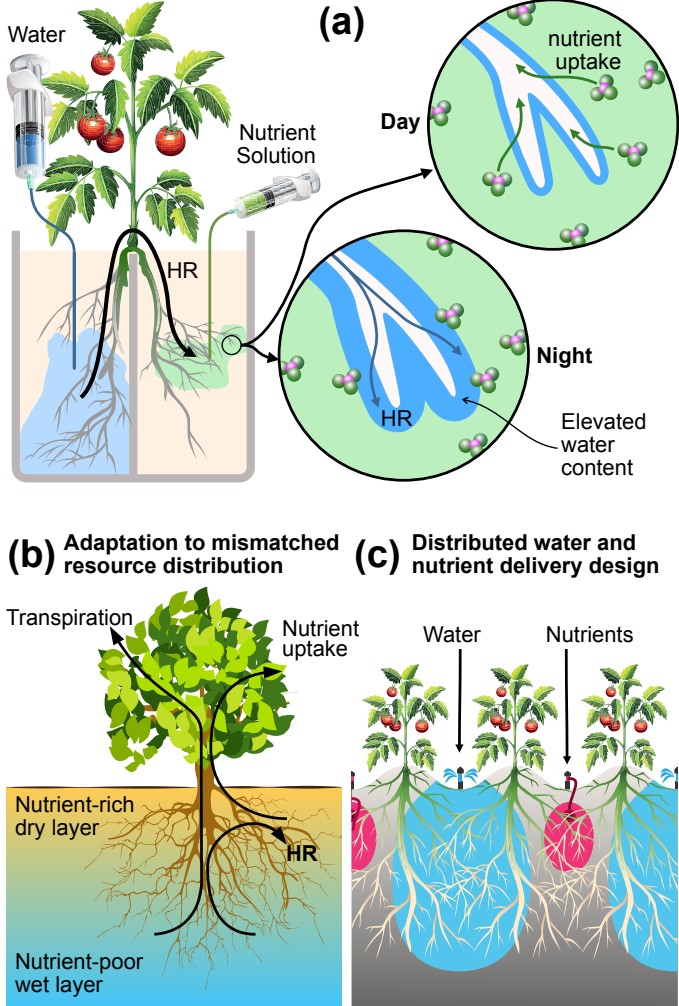

**Figure 8.** Mechanisms, functions, and applications of root uptake under mismatched distributions of water and nutrients in the root zone; (a) schematic representation of how HR supports nutrient uptake under our experimental condition; (b) hypothesized function of HR as an adaptation mechanism in natural systems, where nutrients are concentrated in shallow layers that are prone to frequent drying; and (c) a proposed management practice that can reduce nutrient leaching from irrigated agriculture by capitalizing on the mechanisms elucidated in this study.

## 4 Discussion

The above results directly address the three crucial questions we set out to answer: Does the mismatched distribution of water and nutrients within a soil profile adversely affect plant performance? If not, to what extent are roots able to acquire nutrients from dry soil patches provided that water is available elsewhere? What is the role of HR in nutrient uptake from dry patches?

## 4.1 Aboveground plant performance

We did not observe a measurable difference in reproductive success (number of flowers, fruits, and fruit mass) and nutrient acquisition by the aboveground parts (leaf greenness, nutrient content, and uptake) between any of the treatments, except for a small but significant difference in the mean whole-plant-NDVI between the two control treatments. These results showed that nearly complete separation of water and nutrients does not significantly impact the overall performance of tomato plants. Intermittent irrigation possibly could have alleviated root stress in the dry compartments. Nevertheless, it is noteworthy that the plants subjected to mismatched resource allocation derived all their nutrients from soil patches that undergo pronounced drying conditions without showing any aboveground sign of stress. These soils persistently remained at $-900$ to $-500\,\mathrm{kPa}$ for $85\%$ of the growing period. This moisture status is close to the wilting point, as indicated by the water retention curve (Figure A1). This indistinguishable aboveground performance suggests the existence of a below-ground adaptation mechanism. Our results indicate that the tomato plants subjected to mismatched resource distribution employed strategies of root functions that are distinct from plants grown with matched resource availability. Specifically, we suggest three interacting mechanisms that support nutrient acquisition from dry soil, which are schematically illustrated in Figure 8. Namely, matching root distribution with resource distribution, capitalization on HR water for nutrient acquisition and root support, and HR facilitation via modification of rhizosphere.

## 4.2 Root distributions

The markedly higher concentration of roots in the nutrient-rich dry soil compartment (Figure 4a and 4d) than in the similarly irrigated but nutrient-free compartment indicates that these roots were foraging for nutrients. Due to frequent irrigation and coarse texture of the soils, the highest nutrient leaching to the bottom of the pots likely occurred in treatment **C1**, where nutrients were supplied with $\approx 90\%$ of the irrigation water, followed by treatment **C2**, where nutrients and water were equally divided between the two compartments. Likewise, there was a distinct accumulation of roots at the base of the wet compartment of treatment **C1** (Figure 4b) followed by significant but less pronounced accumulation at the bases of both compartments of treatment **C2** (Figure 4c). The above observation suggests that a substantial proportion of root growth is driven by nutrient availability. It also indicates the existence of root growth regulation mechanisms tied to the sensing of resource availability signals (Bao et al., 2014; Weidlich et al., 2018). If the pots were deeper, it is likely that the roots in the wet compartment of treatment **C1** and both compartments of treatment **C2** would have grown deeper as well. Therefore, our conclusion that nutrient availability dictates root distribution is not necessarily limited to the specific pot dimensions used in this study.

## 4.3 Role of HR

Attributing the water-content and water-potential dynamics reported in Figures 6 and 7, respectively, to HR flux requires eliminating other possible mechanisms that may result in similar fluctuations. In principle, lateral redistribution from relatively moist parts to dry parts of the same compartment can result in a signature that resembles the observed trend. However, this is highly unlikely because all the working five psychrometers across the three replicates recorded synchronized dynamics of

nightly increase and daily decrease in water potential (Figure 7a and Figure S7). Lateral redistribution would require some parts of the soil to lose water at night and gain during the day, which was not recorded by any sensor. Furthermore, the slow

decline in water content (measured by the dielectric sensors) in the dry compartment of treatment **C1** (Figure 6b) suggests that vertical water redistribution is a relatively slow process that does not match the rapid water potential fluctuation in the much drier nutrient-rich compartment of treatment **D**. Finally, the dielectric sensors captured a trace of diurnal pattern in the dry compartment of treatment **D** that is consistent with HR (Figure 6a and Figure S6). But the magnitude of the latter fluctuations is close to the detection limit of the dielectric sensors and could not be relied upon for quantitative analysis.

How can only 10 % of the irrigation water support 60 % of the root growth and be responsible for 100 % of nutrient uptake in treatment **D**? A trivial explanation could be that root growth and nutrient uptake occurred only during the short pulses of nutrient injection. However, the occurrence of HR only in this treatment suggests a substantial role of HR in the adaptation to the mismatched distribution of resources. We propose two possible functions of HR.

First, HR prevents root stress and loss of function by preventing excessive drying (Boyer et al., 2010; Bauerle et al., 2008).

After every weekly nutrient application in treatment **D**, the water content (Figure 6a and 7b) and water potential (Figure 7a) declined rapidly. However, HR allowed the water potential to remain at a stable dynamic equilibrium without ever approaching permanent wilting point $-1500\,\text{kPa}$) (Figure 7a).

Second, HR allows nutrients to remain in solution and be mobile, thereby facilitating nutrient uptake by roots. The role of HR in supporting enhanced nutrient cycling and uptake has been previously noted in field conditions (Bogie et al., 2018;

Cardon et al., 2013; Matimati et al., 2014).

## 4.4 Facilitation of HR

The above observations lead to a critical question: Do roots have agency in regulating HR or their utilization of HR for nutrient uptake is merely capitalization of a "sweet accident", passively governed by physical conditions of the environments (Caldwell et al., 1998; Horton and Hart, 1998; Ryel, 2004). We argue that roots indeed play some role in triggering and regulating HR,

and we submit three complementary evidence to support this argument.

First, HR was detected only in the nutrient-rich dry compartment of treatment **D** but not in the identically wetted nutrient-free dry compartment of treatment **C1**. Roots drive HR by drying the rhizosphere and creating the necessary water potential gradient to pull water from the wet compartment. In addition, extensive root growth provides the necessary flow channel and surface area to carry and release HR water. However, the presence of essential nutrients in the drying soil appears to be an

300 additional condition for HR.

Second, drying also counters HR by dropping hydraulic conductivity of the rhizosphere (van Genuchten, 1980), which restricts the ability of water to diffuse away from root surfaces. Evidence for the role of hydraulic conductivity in controlling HR is indicated by the positive correlation between water potential and HR depicted in Figure 7d. Loss of hydraulic conductance in the soil-plant systems has been previously attributed to the decline in HR magnitude (Meinzer et al., 2004; Prieto et al.,

2010; Scholz et al., 2008). At first glance, this observation appears to contradict the commonplace observation of rapid and spontaneous imbibition during infiltration into dry soils. However, it is crucial to recognize that, unlike the wetting front of

infiltration, the surface of roots during HR typically remains at low water potential and is more susceptible to a drop in hydraulic conductivity (see Appendix A for detailed explanation).

Thirdly, therefore, it appears that modification of that of the rhizosphere that increases hydraulic conductivity would benefit plants by enhancing the benefits of HR. There is a growing consensus on the importance of root hairs for nutrient uptake (Zhang et al., 2018; Bates and Lynch, 2001), which includes that denser and thicker root hairs alter the soil porosity and hydraulic connectivity at the root-soil interfaces (Keyes et al., 2017; Koebernick et al., 2017, 2019). Evidence of this role of root hairs was present in our study, as shown by the pronounced density and thickness of root hairs observed in the nutrient-rich dry compartment of treatment **D** (Figure 5b). Moreover, roots can enhance the rhizosphere's water retention by accumulating rhizodeposits (Carminati et al., 2010; Moradi et al., 2011; Albalasmeh and Ghezzehei, 2014; Ghezzehei and Albalasmeh, 2015). Rhizodeposition released from root tips has been found to decrease the local soil water potential (McCully and Boyer, 1997), facilitate soil aggregation, and is often credited for the facilitation of water and nutrient extraction (Pang et al., 2017; Watt et al., 1994). We observed, albeit in a small scope, organic-coatings of sand in the rhizosphere of the nutrient-rich dry compartment of treatment **D** (see Figure S5), which further support the agency of roots in enhancing HR.

## 4.5    Broader implications of findings

Although our experiments focused on only one plant bred for agricultural purposes under an artificial environment, we can make educated conjectures on how plants can adapt to a mismatched resource environment. The deliberate imposition of extreme separation of resources used in our experiments gives credence to the broader applicability of the proposed mechanisms. The above evidence allows us to propose a conceptual model of how HR plays a central and critical role in plant adaptation to mismatched resource distributions, as illustrated in Figure 8b. The condition presented in this diagram represents a soil profile in an arid or semi-arid environment that experiences frequent and extreme drying of the shallow layers, while soils at depth retain sufficient amounts of water to support transpiration. Moreover, organic matter and plant-available nutrients are preferentially concentrated in the shallow soil. This soil profile resembles most soil natural (Caldwell and Richards, 1989; Cardon et al., 2013; Dawson, 1993) and agricultural (Kizito et al., 2007; Bogie et al., 2018; Wang et al., 2009) field conditions under which HR has been observed. The plant roots are shown tapping both resources by using the HR water to mobilize nutrients that would otherwise remain biologically unavailable.

We also argue that HR is triggered and facilitated to aid in nutrient uptake. The fact that we observed HR in a shallow-rooted herbaceous plant (tomato) suggests that the mismatch of resources rather than climate and plant types are the primary drivers of HR. Increasing water uptake is not necessarily the primary function of HR, as has been postulated elsewhere (Ghezzehei and Albalasmeh, 2015; Carminati et al., 2016; Meinzer et al., 2004). Indeed, our observations of root-zone water dynamics (Figure 7b) suggest that the net contribution of HR to transpiration at the whole-plant scale is small. In our experiment, nutrients were delivered with low weekly pulses of irrigation. In real conditions, precipitation events in arid and semi-arid regions can be less frequent. Therefore, our conceptual model suggests that HR is critical for the sustenance root functions during extended drought spells.

Furthermore, maintaining the soil water status above a detrimental threshold via HR would permit soil microbes to carry out essential nutrient cycling functions in the rhizosphere. Nutrient cycling was likely less critical in the current study because the sand lacked organic matter, and all nutrients were delivered in a plant-available form. However, the effects of HR on microbial functions in the shallow dry soil layers can be substantial under field conditions (Cardon et al., 2013), where the bulk of plant nutrients are likely to exist in non-available form affixed to mineral surfaces and as part of organic matter (Keiluweit et al., 345 2015; Li et al., 2004). The HR-facilitated microbial activities can be extensive at the ecosystem scale through interaction with the rhizodeposition dynamics (Williams and de Vries, 2020). This hypothesis is consistent with the frequent occurrence of HR in deep-rooted shrubs of arid and semi-arid regions (Kizito et al., 2007; Bogie et al., 2018).

## 4.6   Implication to sustainable agriculture

In this study, we used relatively shallow and closed pots to eliminate differences in total nutrient and water availability. How-
350 ever, had the bases of the pots been open or deeper, a fraction of the nutrient supply could have leached below the rooting depth in the control treatments **C1** and **C2**. Higher root density at the bases of the pots in these treatments suggests that the roots were able to utilize the leached nutrients. This finding highlights a persistent curse of modern irrigated agriculture, in which a substantial fraction of applied fertilizers leach below the rooting depth (Bowles et al., 2018). As a result, while the NUE in many industrialized countries has been increasing at a modest rate, the yield gains achieved in most developing countries
over the past half-century came at a significant decline in NUE (Zhang et al., 2015) and environmental and ecological degra-dations, including air and water pollution and the accumulation of potent greenhouse gases (Bowles et al., 2018; Balmford et al., 2018). To meet the 2050 global food demand while safeguarding environmental quality would require harvested N to increase by $45\%$ while NUE to increase from $40\%$ to $70\%$ (Zhang et al., 2015). Our findings suggest that the colocation of nutrients and water, which is the main driver for N loss by leaching and volatilization (Bowles et al., 2018), is not necessary
to maintain productivity. Thus, spatially isolating the bulk of irrigation water from the applied N can be effective in drasti-cally cutting N losses. In Figure 8c, we propose one such approach. The scheme involves meeting the transpiration demand of crops by irrigating every other row while using the rest of the rows for delivering nutrients in small quantities. The approach resembles a well-established partial-root-drying (PRD) method of irrigation method practiced in arid regions, including Israel and Australia (Bielorai, 1982; Dry et al., 2000). In PRD, every other row is irrigated in an alternating schedule, whereas our
proposed scheme requires that nutrients and water be delivered to dedicated rows. This proposal is consistent with a recent recommendation by Vetterlein et al. 2020 to utilize the integrative function of plant decision-making and self-regulation the sustainable management of agricultural systems.

## 5   Conclusions

Our findings demonstrated that tomato plants can utilize heterogeneously distributed resources without adverse impact on their
performance. Specifically, we showed the ability of plants to acquire $100\%$ of their nutrient needs under the extreme mismatch of water and nutrient distributions. We provided evidence that suggests a successful adaptation to such an environment involves

coordination between components of the root system that inhabit environments with contrasting resource availability. Critical to this mechanism is a reliance on multiple strategies, including extensive root proliferation that allows rapid nutrient capture from immediate widows of availability under favorable moisture conditions, and sustained HR to support an active root system and facilitate nutrient transport under unfavorable or drought stress conditions. It appears that the overall plant nutrient demands drive the occurrence of HR because of HR's role in supporting the acquisition of nutrients. Regulation mechanisms that control HR occurrence and magnitude include root adaptions at different spatial and temporal scales, including the extensive proliferation of root branches, thick root hairs, and rhizodeposition formation. Finally, we provided a conceptual model of how HR may play an integral in plant adaptation to mismatched resource distributions and suggested a nature-inspired irrigation scheme to minimize nutrient losses and environmental pollution.

*Code and data availability.* The data sets and R code were uploaded in Dryad with a doi:10.6071/M39M2T. The unpublished dataset and code for review was shared using this temporary link: click for data sets and code for review.

## Appendix A:  Effect of wetness on HR

The pattern of HR reported in Figure 7d appears to contradict analogy with infiltration, where flux density is expected to rise with decrease in initial wetness. The observed pattern can be explained assuming the water flux at the root-soil interface is governed by a flux law that is analogous to Buckingham-Darcy law

$$q = \frac{\psi_r - \psi}{\delta} K(\psi)$$

The first term denotes the water potential gradient, i.e., difference between the value at the root surface ($\psi_r$) and the rhizosphere ($\psi$), $\delta$ denotes the thickness of the rhizosphere, and $K$ is hydraulic conductivity of the rhizosphere. The relationship between $\psi$ and HR flux depends on both hydraulic conductivity and hydraulic gradient. Notice that drying of the rhizosphere soil affects these two factors in opposite directions–by increasing the gradient and decreasing the conductivity. Therefore, the net effect should be dependent on the relative magnitudes of these effects. For simplicity, if we assume that $K$ is described by $K = K_s e^{\alpha(\psi - \psi_0)}$ based on Gardner 1958, where $K_s$ is the saturated hydraulic conductivity, $\psi_0$ is air-entry water potential, and $\alpha$ is a fitting parameter related to soil pore size characteristics. Then, the above flux law can be simplified as

$$\frac{\delta q}{K_s} = (\psi_r - \psi)e^{\alpha(\psi - \psi_0)}$$

It can be shown that the above scaled flux has a maxima at $\psi^* = (\alpha\psi_r - 1)/\alpha$. Above this threshold ($\psi > \psi^*$), flux increases with drying of the rhizosphere, while below the threshold ($\psi < \psi^*$) the opposite would occur. In the range of measurements observed in this study ($-1000$ kPa $\leq \psi \leq -100$ kPa), the latter appears to dominate.

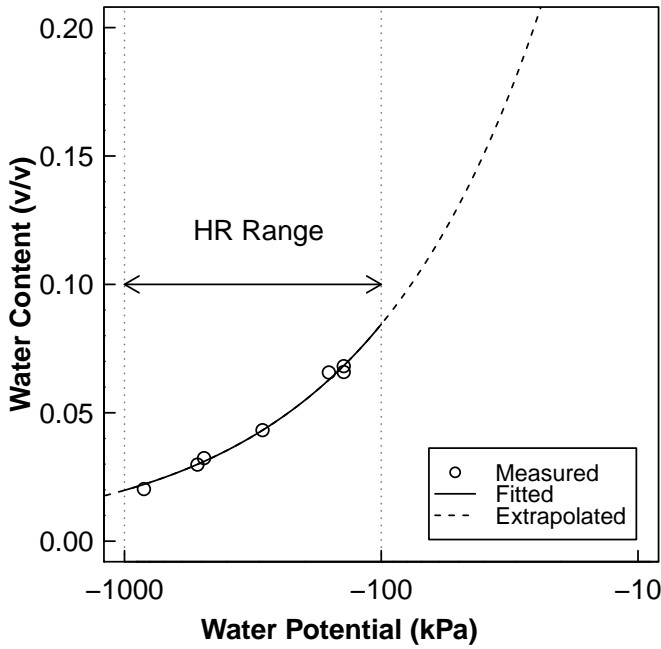

**Figure A1.** Water retention as a function of water potential derived from independent soil characterization. The points represent the measurement of water content and water potential using the potentiometer, and the solid line is the best fit Brooks Corey's model. The measurement points adequately cover the range of water potential at which HR was observed using psychrometers. This curves was used for conversion of water potential to water content and calculation of HR water flux shown in Figure7b, c and d. The fitted parameters of the curve are air-entry water potential ($\psi_0 : -6.51$ kPa) and the dimensionless shape factor ($\lambda : -0.63$). The saturated water content was estimated from the bulk density as saturated water content,$\theta_s : 0.47$ v/v and that residual water content was assumed zero.

**Table A1.** The mean, standard deviation of physiological indicators, and the p-value of Welch's ANOVA test across treatments. Note: comparison of Leaf NDVI was performed both at the 3rd to 6th branches (equivalent to the normalized plant height of 0.8 to 0.9 ) and the whole plant scale. Values with different letters indicate significant difference ($p < 0.05$).

| Variables | Treatments | | | $p$ value |
|---|---|---|---|---|
| | D | C1 | C2 | |
| Total dry mass (g) | $6.23 \pm 0.41$ | $6.57 \pm 1.01$ | $6.84 \pm 0.34$ | 0.30 |
| Shoot dry mass (g) | $5.37 \pm 0.54$ | $5.87 \pm 0.87$ | $6.19 \pm 0.43$ | 0.28 |
| Initial dry mass (g) | $1.43 \pm 0.34$ | $1.26 \pm 0.02$ | $1.05 \pm 0.12$ | 0.16 |
| Flower no. | $3.67 \pm 2.08$ | $4.00 \pm 2.65$ | $2.67 \pm 1.53$ | 0.73 |
| Fruit no. | $2.00 \pm 1.00$ | $1.67 \pm 1.53$ | $2.00 \pm 1.00$ | 0.95 |
| Fruit dry mass (g) | $0.85 \pm 0.13$ | $0.70 \pm 0.61$ | $0.65 \pm 0.36$ | 0.70 |
| Fruit N content (%) | $2.23 \pm 0.21$ | $1.36 \pm 1.23$ | $1.87 \pm 0.09$ | 0.28 |
| Fruit N uptake (mgN) | $19.21 \pm 4.9$ | $14.37 \pm 13.57$ | $12.22 \pm 6.8$ | 0.48 |
| Shoot N content (%) | $1.35 \pm 0.10$ | $1.32 \pm 0.11$ | $1.16 \pm 0.21$ | 0.21 |
| Shoot N uptake (mgN) | $69.67 \pm 6.11$ | $80.11 \pm 6.65$ | $76.32 \pm 8.15$ | 0.28 |
| Total N uptake (mgN) | $70.73 \pm 6.71$ | $78.63 \pm 6.42$ | $77.01 \pm 2.94$ | 0.43 |
| N usage efficiency (%) | $59.04 \pm 5.60$ | $65.63 \pm 5.36$ | $64.28 \pm 2.45$ | 0.43 |
| Leaf NDVI $(0.8 - 0.9)$ | $0.88 \pm 0.01$ | $0.86 \pm 0.04$ | $0.89 \pm 0.01$ | 0.10 |
| Leaf NDVI (whole plant) | $0.84 \pm 0.10$ ab | $0.82 \pm 0.06$ b | $0.86 \pm 0.05$ a | $< 0.001$ |

**Table A2.** The elemental composition of essential macro- and micro-nutrients in the irrigating nutrient solution. Note: the elemental concentration was reported as the normalized concentration to the nitrogen level. The calculated results were based on the information from the product manufacture label.

| Macro-<br>& Micro-Nutrients | Normalized<br>Concentration |
|---|---|
| Nitrogen | 1.00 |
| Phosphorus | 0.46 |
| Potassium | 1.45 |
| Calcium | 0.55 |
| Magnesium | 0.15 |
| Sulfur | 0.18 |
| Boron | < 0.01 |
| Copper | < 0.01 |
| Iron | 0.01 |
| Manganese | 0.01 |
| Molybdenum | < 0.01 |
| Zinc | < 0.01 |

Note: commercial hydroponic nutrient solution
(General Hydroponics, Santa Rosa, CA) derived from
Ammonium Nitrate, Calcium Nitrate, Magnesium
Nitrate, Magnesium Sulfate, Monopotassium
Phosphate, Potassium Nitrate, Potassium Sulfate,
Sodium Molybdate was diluted accordingly for
nutrient application.

*Author contributions.* J.Y., T.A.G. designed and performed the research; J.Y., N.A.B, and T.A.G. analyzed the results and wrote the paper.

*Competing interests.* The authors declare no conflict of interest.

*Acknowledgements.* We thank Luis Davila, Lythia Meza, Yulissa Perez Rojas, Mimi Pomephimkham, and Joseph Veneracion for assistance
with soil and plant sampling and characterization. Assistance from Miguel Manansala, Kennedy Nguyen (Microscopy & Imaging Facility), Liying Zhao (Environmental Analytical Lab) and Justin Van De Velde (Stable Isotope Lab) is gratefully acknowledged. This study was supported by USDA-NIFA Grant #2016-67019-25283.

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
