# Peer review of "Root uptake under mismatched distributions of water and nutrients in the root zone."

_Biogeosciences, 2020_

## Referee Comment (RC1) · Anonymous Referee #1 · 2 Jun 2020

The manuscript entitled "Root uptake under mismatched distributions of water and nutrients in the root zone" aims to test how mismatched distribution of water and nutrient influence nitrogen acquisition and plant growth. The authors further investigate how hydraulic redistribution and changes in root morphology can explain their results. While the objective of the study is very relevant and rather clearly defined and justified, the rest of the manuscript (material and methods, results, discussion and conclusion) is hard to follow, with crucial elements lacking from the material and methods. It makes it difficult to understand why the authors did some measurements and what they really measured. In the discussion and conclusion, I found some part too speculative. For example, how could you conclude so strongly about the crucial role of root hairs and production of mucilage, only based on non-quantitative microscopic observations? I

suggest you better describe what you really demonstrated and what your results only suggest. Overall, I think that the data provided here are of good quality, that the design was well though, but the manuscript is poorly written. See specific comments to help you to improve it.

Abstract:

Please precise which plant (or at least type of plant) you grew as I am not sure that trees, herbs and grass shows the same adaptations to mismatches. At least, it should be proven before concluding it. We lack the experimental design (at least briefly mentioned) in the abstract

l.13 -15 : It is too strong from my perspective. You did not quantified root hair density, neither production of root mucilage.

Introduction:

l. 28 -31 : I was pleased to read that you mention the role of rhizospheric soil microbes to make nutrients available for plants. This could, and for my perspective should, be mentioned in the discussion too (although not too extensively as you did not measured any microbial parameter here).

You mention specific adaptations of plants to water or nutrient deficiency (or heterogeneous distribution), namely: (i) Preferential growth in moist areas and modifications of root exudation (l.32-36) and hydraulic redistribution (l. 38 – 42). In these two paragraphs, you develop more adaptations to water scarcity or heterogeneity in fact. Adaptations to N deficiency or heterogeneity are less developed. For example, roots of a non-legume plant can forage toward the roots of a legume plant (Weidlich et al., 2018). Associations with soil microbes, such as N-fixing bacteria and mycorrhizae are as well strategies to enhance N acquisition and avoid growth limitations. Differences in root morphology (SRL, ratio root length/dry mass) of absorptive roots are typically used to describe foraging behavior of roots to acquire root N (a mobile nutrient). Proliferation

of root hairs (which is not mentioned here, although it seems to be important for your article), or root clusters (highly branched roots) are more known to enhance acquisition of P, a less mobile element often found in patches (Lambers et al., 2011; Bates et al., 2001). With regards to adaptations of roots to water scarcity, see as well the recent article from Bristiel et al., (2019). The adaptations cited here do not sufficiently cover the topic.

Weidlich, E. W., Temperton, V. M., & Faget, M. (2018). Neighbourhood stories: role of neighbour identity, spatial location and order of arrival in legume and non-legume initial interactions. Plant and Soil, 424(1-2), 171-182.

Lambers, H., Finnegan, P. M., Laliberté, E., Pearse, S. J., Ryan, M. H., Shane, M. W., & Veneklaas, E. J. (2011). Phosphorus nutrition of Proteaceae in severely phosphorus-impoverished soils: are there lessons to be learned for future crops?. Plant Physiology, 156(3), 1058-1066.

Bates, T. R., & Lynch, J. P. (2001). Root hairs confer a competitive advantage under low phosphorus availability. Plant and Soil, 236(2), 243-250.

Bristiel, P., Roumet, C., Violle, C., & Volaire, F. (2019). Coping with drought: root trait variability within the perennial grass Dactylis glomerata captures a trade-off between dehydration avoidance and dehydration tolerance. Plant and soil, 434(1-2), 327-342.

l. 50 – 53: While the objective was rather clearly described, I do not see the point with these last sentences.

Material and methods

In general, this section lack clarity and there is several important missing information. The methods are often described without explaining their aim. The subsection 2.1 (which could be renamed experimental design) lack to present the experimental design. Instead, the signification of treatment D, C1 and C2 is given at the beginning of the results! I can't find figure S1. I lack as well the number of replicates. The duration

of the experiment should be given here too. The quantities of N, water, how are loss compensated, where it is added should be described... please report what was done with accuracy.

l.62 – 67: the measurement of water content and water potential belong to plant and soil characterization

l. 80: Please define NUE, I guess this is nitrogen use efficiency, but this should be written.

l. 86: What do you mean by "further gravimetric measurements"?

l.88-93: It is not clear why you are doing these microscopic analyses. Why laser of two different wavelengths are used? What is gold coating for?

Results

The subsections are confusing. Is plant water and nutrient uptake (3.3) not related to plant physiology (3.1)? Please reorganize. Moreover, some parts belong to material and methods, other to discussion. Focus on what you have observed here.

l.105 – 115: This belongs to material and methods.

l. 118 -120: This is your interpretation of the results. It should go to discussion.

l.122 -124: This belongs to introduction

l. 127: How did you test that root density do not differ between the two compartments? By comparing root masses? If this is the case, it is thus not root density but root mass. Moreover, in table A2, the wet and dry compartments of the treatment D are significantly different.

l. 127- 128 : this belongs to material and methods

l. 130-131: Belongs to material and methods

l.131- 134: Please indicate what this higher root masses in the deeper part suggests

in the discussion. Here you should describe the results.

l.135: Again root density or root mass?

l.136- 138: again, belong to discussion. Moreover, avoid detailing twice the same idea. An increase in root mass in the deeper layer is seen in the three treatments D, C1 and C2.

l. 138- 140: This should be stated in material and method, not here.

l.140: Did you measure root growth? Or are you indicating root mass? Root mass is not equal to root growth as the root mass at a given point depends on root growth, and root death (life span / root turnover).

l.143. 145 from "which is…" belongs to discussion.

l.146: How did you measure root hair density? What test did you do to conclude for significant differences?

l. 147 – 148: This belongs to discussion

l.150: Avoid starting a new paragraph with "the above observations". It suggest you are still developing previous ideas, so why starting a new subsection?

l. 151 – 153: Belongs to introduction

l.155: Did you describe the frequency of the irrigation?

l. 157: The information about the frequency of N addition should be given in material and method.

l.161: How did you converted soil water potential data to rhizosphere water content?

l. 163-164: Avoid opinion terms such as " closer inspection".

l.166: Do no cite literature in the results, you should describe what you found here.

l. 168: What do you mean by "habitable environment"? For the roots? For rhizospheric

microbes? Your focus here is not nutrient uptake, please stay stick to it.

l. 173 -175: Again, this is not the description of the results.

l. 176 – 179: This belongs to discussion

l. 180: Do you assume that the organic coating is root mucilage? How did you quantified it? What are the two fluorescent wavelength for?

L.181: keep suggestion to the discussion

l. 181: This is an interpretation, not a result.

Discussion

l. 184 – 186: This belongs to introduction

l.192: How could you confidently conclude that plant performance are less sensitive to localized scarcity in water and N if nutrient and water are sufficient in other locations where the roots forage. You did not tested it. To know it you should have a mismatched distribution of water and nutrients, with an overall limitation in water and N (compared to your treatment D).

l. 194: I can't see what allow you to draw this conclusion here. Nothing written in the paragraph above allow to conclude it, although I think that you are right to point different plant strategies in case of mismatches.

l. 197 – 198 : Sentence not clear

l. 198 : You did not measure root proliferation as far as I have understood and what do you mean by this term: root growth? Root turnover?

L.199: What is multi-scale signaling and feedback? This is too vague.

l. 200: You did not describe root allocation in the results. You surely want to say that this is the relative mass of roots in the two compartments? Or in the various depths? Please specify it. I can't see how it points a whole plant scale regulation of root growth.

Please explain.

l. 201: This confirms the foraging behavior of non legume roots to legume roots (Weidlich et al. 2018).

l. 204 – 205: This is one of the most interesting result of the study. Please detail more.

l. 206: What do you mean by "vigorous"? How did you measure it? It is not clear to me how drying after wetting event can indicate vigor.

l.214: This is an important result too.

l. 218 -219: What do the references refer to? You conclude here from your own results and cite the related figure. I guess the references indicate that this has been previously shown?

l.221: Need a reference

l. 226 – 229: Avoid finishing with limitations. Specify them either in the conclusion or in the discussion but not at the end as this is the last take home message for the reader.

Conclusion:

L.231: "could" or "did"? Be clear with what you have demonstrated. In general, better differentiate what you showed and what your results suggests.

l. 243: How did you measure root activity?

l. 244: What is a vigorous nutrient cycling. Did you measure it?

l. 250-260: I enjoyed the final thought about application, but it makes the conclusion quite long and bring new ideas. This paragraph may be moved to the discussion.

Table A1: I would enjoy a graph or table with the values measured here. N uptake is central in your article (according to the objectives).

Figure 3: What does the different color means? It would be better to rename treatments

with an easy understandable name, instead of D, C1 and C2, which looks more a code for labeling pots.

---

## Referee Comment (RC2) · Anonymous Referee #2 · 10 Jun 2020

I was generally pleased to read through the manuscript titled Root uptake under mismatched distributions of water and nutrients in the root zone. The manuscript sets out to demonstrate that plant roots are still able to operate well under conditions where water and nutrients are partitioned in segregated regions. The article presents a novel and imaginative set up well within the capacity to monitor a vast array of soil and plant physical and philological features. Results for the most part are clear and concise and the writing is very comprehendible. With the praise being said, there are a few critical points that need to be addressed in the manuscript. Most of the points pertain to organization, but a few are on the science itself. The figures are in a strange order. I think the first figure that's referenced is the last figure in the manuscript, and this makes no sense. There are a lot of subfigures that are never mentioned or mentioned in a strange

order. I've gone through and make marks regarding these points and recommend that the authors make amendments accordingly. Just have the figures appear as they are mentioned in the text and make sure to mention all of the figures that you are presenting. That is pretty simple. More pertinent is the matter of the science. In particular, the focus of the study somewhat diverges and tries to come back together towards the end of the manuscript. In the beginning, the authors are describing this split column root growth experiment in the context of nutrient acquisition and plant development. The authors then attempt to push the notion of hydraulic redistribution in the results later on. It comes off as a bit shoehorned in. My particular issue with this is that it is that your results might be suggesting a very subtle and highly local redistribution of water. The authors then demonstrate that HR is actually less effective under drier conditions when it would be most needed. Towards the end, I was almost convinced that HR wasn't a relevant topic matter. However, the authors did manage to sway me back in slightly when they were trying to make the argument that it was used for the nutrient uptake. I think the authors need to really focus on the subtlety that they are highlighting with their results and draw from some more fundamental principles to base their arguments. Consider that root nutrient acquisition relies on enzymatic reactions that may require a sufficient quantity of water to enact. I think something simple but fundamental would give this study a stronger foundation for its claims. The study already does a good job of illustrating that roots are not just passively behaving below ground. Their ability to actively take up water and nutrients is already interesting. The authors just have to better reconcile the results in figure 3 a and figure 4 b. They appear contradictory.

My specific comments can be found in the attached annotated manuscript document.

Please also note the supplement to this comment:
https://www.biogeosciences-discuss.net/bg-2020-109/bg-2020-109-RC2-supplement.pdf

**Supplement:**

[revised manuscript text omitted]

---

## Referee Comment (RC3) · Anonymous Referee #3 · 12 Jun 2020

This manuscript presents experimental evidence that plants can satisfy their water and nutrient demand from mismatchingly distributed water and nutrient resources, if the overall available amount is sufficient. The plant adaptation strategies and regulating mechanisms related to this are discussed. Overall, this is a well-designed contribution of high interest. However, the methods in part lack clarity, and the results and discussion are in parts too speculative. My first two points are about nomenclature: Comment 1: The first is the definition of the term rhizosphere. There are different ways in literature how to use the term rhizosphere and thus I think it is important to define clearly what this term means in this paper. I think this paper rather means a part of soil which has a high root density, i.e. it is more used in the meaning of "root zone". That could be confusing as a lot of other work understands the rhizosphere much more locally in

form of gradients in the concentration of root-influenced solutes or other compounds extending from the root surface to the 'bulk' soil (Darrah et al., EJSS 57, 2006). Comment 2: The second is the term "exudates". It is often used quite differently in different papers. I rather tend to distinguish "root exudates" as low-molecular weight organic carbon (such as citrate, sugars) and mucilage. An overarching term that includes both exudates and mucilage would be "rhizodeposition" (Oburger and Jones, Rhizosphere 6, 2018). I encourage the authors to also use this nomenclature.

Some methodological aspects were also not clear to me: Comment 3: I could not find in which depths the water potential sensors were installed. I could also not infer in how far it is justified to call the resulting value a "rhizosphere" water potential. Is it not rather the water potential in the soil layer that has the highest root length density? One could understand this from your sentence on page 3, line 64: "...to measure the water potential of the root zone". Comparing the water content that was computed from the rhizosphere water potential (Fig. S5c) with the water content that was measured with the dielectic water content sensor that was installed in the middle of the compartment (Fig. S3a), I can hardly see a difference. Comment 4: How can you be sure that the water increase in the root zone with highest root length density results from HR? Root water uptake and injections will create water potential gradients within one compartment that could result in redistribution of water in the soil.

Comment 5: The structure of the paper needs attention. I suggest, for example, to move the paragraph lines 105-115 page 4 to the description of the split-root experiment in the Methods section. Then, the "D" and "C1" will be easier to understand in line 64 on page 3.

Some claimed results seem a bit too speculative to me: Comment 6: The reason for root accumulation at the bottom could also just be that the pot was too short. I.e., if almost all the carbon in C1 is invested in the wet and nutrient rich compartment, it may be possible that the roots would have grown much deeper than in the other treatments if they had been given the space. Comment 6: "Moreover, multi-scale signalling and

feedbacks appear to be involved": How could you support this statement with your results? Comment 7: While it is known that hormonal signalling may regulate the transpiration demand at the leaves, the water flow into our out (HR) of the roots follows (passively) local hydraulic gradients between xylem and soil (e.g. Rothfuss and Javaux, Biogeosciences 14, 2017). What regulation mechanisms exactly do you mean by your statement "HR is biologically-mediated"? Would that be regulation of root hydraulic properties? How could you support that with your results?

Minor comments - I could not see that the number of replicates was mentioned in the Methods section. P3 L57: How long did it take the plants to reach that height? P3 L59: How many roots were there at this stage? Was the tap root recognisable and was there a strategy to place it into a specific compartment? P3 L85: When you scooped out the soil, did you cut the roots within these 2cm intervals? Fig. 2a: I do not see the relevance of Fig. 2a. I also suggest to split the rhizosheath and root mass distribution to two separate figures. Fig. 2h: dry and wet labels are confusing for this treatment. P8 L165: "taken up by the roots" P11 L211: The absence of HR in C1 was not mentioned in the Results section.

―――――――――――――――――――――

---

## Author Comment (AC2) · 21 Jul 2020

I was generally pleased to read through the manuscript titled Root uptake under mismatched distributions of water and nutrients in the root zone. The manuscript sets out to demonstrate that plant roots are still able to operate well under conditions where water and nutrients are partitioned in segregated regions. The article presents a novel and imaginative set up well within the capacity to monitor a vast array of soil and plant physical and philological features. Results for the most part are clear and concise and the writing is very comprehendible. With the praise being said, there are a few critical points that need to be addressed in the manuscript.

Most of the points pertain to organization, but a few are on the science itself. The figures are in a strange order. I think the first figure that's referenced is the last

figure in the manuscript, and this makes no sense. There are a lot of subfigures that are never mentioned or mentioned in a strange order. I've gone through and make marks regarding these points and recommend that the authors make amendments accordingly. Just have the figures appear as they are mentioned in the text and make sure to mention all of the figures that you are presenting. That is pretty simple.

More pertinent is the matter of the science. In particular, the focus of the study somewhat diverges and tries to come back together towards the end of the manuscript. In the beginning, the authors are describing this split column root growth experiment in the context of nutrient acquisition and plant development. The authors then attempt to push the notion of hydraulic redistribution in the results later on. It comes off as a bit shoehorned in. My particular issue with this is that it is that your results might be suggesting a very subtle and highly local redistribution of water. The authors then demonstrate that HR is actually less effective under drier conditions when it would be most needed. Towards the end, I was almost convinced that HR wasn't a relevant topic matter. However, the authors did manage to sway me back in slightly when they were trying to make the argument that it was used for the nutrient uptake. I think the authors need to really focus on the subtlety that they are highlighting with their results and draw from some more fundamental principles to base their arguments. Consider that root nutrient acquisition relies on enzymatic reactions that may require a sufficient quantity of water to enact. I think something simple but fundamental would give this study a stronger foundation for its claims. The study already does a good job of illustrating that roots are not just passively behaving belowground. Their ability to actively take up water and nutrients is already interesting. The authors just have to better reconcile the results in figure 3 a and fgure4b. They appear contradictory.

> **Response**: We thank the reviewer for the thorough and critical review. The criticism about order of figures and logical flow of content was shared by the other reviewers. In response, we have made substantial reorganization

of the content. We revised the figure that describes the experimental design and moved it to the main body of the manuscript as Figure 1. It now includes clear definition of the abbreviations, sensor placement and dimensions of the pot, which will make the results and discussions easy to follow. All the figures now appear in the order they are referred. All subfigures are now described in the body of the manuscript.

The revised manuscript is now more streamlined in a manner that the reader can easily follow to get to the main points of this paper. The first point we make is that nearly complete separation of water and nutrients did not significantly impact the overall performance of the tomato plants we investigated. This finding is the basis for the key question addressed in this paper: *Given that plants are not underperforming in this non-ideal resource availability, what are the adaptation mechanisms used by plants to survive and thrive?* Our answer has multiple related parts. First, there is a high-density root growth in the nutrient-rich dry soil (treatment **D**) compared to the nutrient-poor dry soil (treatment **C1**). Moreover, the roots in the nutrient-rich dry compartment were concentrated in the mid-section, where the nutrient pulses were applied. In contrast, when nutrients are added with the bulk water (in both versions of the control treatments), the highest root density was observed at the bottom of the pots, likely because the nutrients were leached down with the water. These observations support that (a) the roots can acquire nutrients from dry nutrient-rich soil and (b) root density is in part correlated with nutrient concentration. Moreover, the SEM images suggest that there is higher density of thicker root-hairs (qualitative observation) in the nutrient-rich dry soil compared to nutrient-rich-wet and nutrient-poor-dry compartments. This is consistent with the literature that describes the importance of root-hairs in nutrient acquisition (Bates, Lynch, 2001; Zhang et al., 2018).

The above observations lead to a more specific question: *how are these roots surviving and growing in the dry environment and how are they able to mobilize the nutrients?* Our psychrometer data suggests that a significant increase in water potential at night time. We interpreted this as hydraulic-redistribution from the wet compartment. We rued out internal redistribution of water within the dry compartment as none of the sensors placed there detected an out-of-phase dynamics. We argue that this wetting is responsible for supporting root proliferation and mobilization of nutrients. As the reviewer stated, the wetting can also be important in increasing microbial and enzymatic activity that is essential for nutrient acquisition. However, the latter role of hydraulic-redistribution was likely to be minimal under our experimental condition because (a) the soils lack organic matter and (b) all the nutrients were provided in plant-available form. That said, it is likely that in field conditions hydraulic-redistribution plays an important role in fueling enzymatic and microbial functions.

All the figures are reformatted for consistency and the key findings have been moved to the main body. The reviewer stated "...better reconcile the results in Figure 3a and Figure 4b. They appear contradictory." Both these figures show the water content dynamics in the nutrient-rich dry compartment. The data in Figure 4b (now appears as Figure 7b) was indirectly calculated from the matric potential data measured by thermo-couple psychrometers. We used soil-specific water retention curve for the conversion. Because the psychrometer has much higher sensitivity to small changes in water potential (hence, water content), the data pattern reveals more pronounced dynamics. The data in Figure 3a (now appears as Figure 6a) is derived from direct measurement using dielectric moisture sensor. The sensitivity of the dielectric sensors is not sufficient to fully

reveal the hydraulic-redistribution dynamics.

Responses to specific comments are provided blow each comment. To facilitate review of our responses, we added all the figures at the end of this document. We added three figures during this revision. Most figures have been revised and the captions have been expanded and clarified.

**1  Specific Comments**

https://www.biogeosciences-discuss.net/bg-2020-109/bg-2020-109-RC2-supplement.pdf The reviewer provided several comments on an annotated manuscript. All identified errors were fixed. Recommendations on wording were mostly accepted as suggested or replaced with alternative language that improved clarity. Substantial specific comments are addressed below. We used the line numbers of the original manuscript to refer to each question. The corresponding responses are provided under each comment.

1. L1-16: The reviewer added these comments and suggestions for clarifying the abstract 'Split in what way? Perhaps you could describe this here in a bit more detail', 'I think this sentence isn't needed. It's vague, unclear, and nicely explained in the subsequent text.', 'It almost reads like two abstracts. The relevance of HR seems like the specific phenomena that the manuscript is working on. Perhaps it's best to just focus more on this as to not distract the readers.', and 'Brief pulses of what?'
   **Response**: The abstract was rewritten to address these comments and the comments of the other reviewers: "Most plants derive their water and nutrient needs from soils, where the resources are often scarce, patchy, and ephemeral. It is

not uncommon for plant roots to encounter mismatched patches of water-rich and nutrient-rich regions in natural environments. Such an uneven distribution of resources necessitates plants to rely on strategies to explore and acquire nutrients from relatively dry patches. We conducted a laboratory study that elucidates the biophysical mechanisms that enable this adaptation. The roots of tomato seedlings were laterally split and grown in two adjacent, hydraulically-disconnected pots, which permitted precise control of water and nutrient applications to each zone. We observed that physical separation of water-rich and nutrient-rich zones does not significantly hamper plant productivity. Specifically, we showed that soil dryness does not reduce nutrient uptake, provided that the whole plant has access to sufficient water elsewhere in the root zone. We identified localized root proliferation in nutrient-rich dry soil patches as a critical strategy that enabled nutrient capture. Furthermore, high-frequency water potential measurements revealed nocturnal rewetting of the nutrient-rich but dry soil compartments. We interpreted this as a water-potential-gradient driven transfer of water from the wet to dry compartments, a process commonly known as hydraulic redistribution (HR). The occurrence of HR prevents the nutrient-rich soil from drying to the level of permanent wilting and subsequent decline of root functions. Moreover, cyclic rewetting of the rhizosphere likely increases nutrient mobility and uptake. It is also possible that roots facilitate HR by increasing root-hair density and length and deposition of organic coatings that increase water retention. Therefore, we conclude adaptation to mismatched resource distributions calls for plant-controlled biophysical regulation of soil and water dynamics in the rhizosphere. Our findings support a nature-inspired nutrient management strategy for significantly curtailing water pollution from intensive agricultural systems."

2. L24: In general, this does sound intuitively correct. However, doesn't this also rely on soil texture?
   **Response:** We agree with the reviewer that soil texture is an important factor that

regulates rate of evaporation and soil drying. In field conditions, plants are likely to experience mismatched distribution of resources when nutrient-rich surface layers dry faster and more often than nutrient-poor subsurface layers. This is likely to occur in coarse textured soils. In fact, most of the studies that reported field observation of hydraulic redistribution were carried out in areas with coarse soil texture (Neumann, Cardon, 2012). We added the following sentence. "This effect is likely to be pronounced in coarse textured soils that dominate most arid and semi-arid soil soils (Rodell et al., 2004)"

3. L33: "tracks" water infiltration patterns? Does this mean roots follow preferential paths of water infiltration?
**Response:** We rephrased it to be consistent with the cited source: "In water-limited areas, rooting depth generally coincides with infiltration depth (Fan et al., 2017).

4. L34: Provide a bit of mechanistic detail. How are plants able to do this?
**Response:** Changed to
"Locally, roots can also respond by increasing the water retention capability of their immediate surroundings (the rhizosphere) by releasing a cocktail of organic compounds that sorb water and promote soil aggregation."

5. L36: Again, what are some of the mechanisms that facilitate this? McKay Fletcher et al. 2020 recently proposed citrate enhanced uptake on the basis of exudation.
**Response:** We revised the sentence as
"Furthermore, root exudation (release of low-molecular-weight rhizodeposits) can increase nutrient availability and accessibility by freeing tightly-bound nutrients (e.g., (McKay Fletcher et al., 2020)) and priming microbial mineralization of nutrients (e.g., (Keiluweit et al., 2015))."

6. L40: Be more specific as to where the claim is coming from. "Studies have found

that..." or "Reports have shown...". We don't want to just take suggestions from people in line at the grocery store (however wise they might be). :) I'm not quite sure if I'm following the rationale here. HR is water that is initially taken up by roots and subsequently released in drier regions of soil. How and/or why should the nutrients also be carried and released? I was of the understanding that plants take up nutrients via enzymatic reactions. I don't see how or why these nutrients would be re-located in the same way that water is. I could understand the water taken up and re-released in dry patches, which helps plants to take up nutrients in those dry patches, but I struggle to understand a sort of nutrient lift or nutrient redistribution. Perhaps I've just misunderstood the statement. If so, could you clarify what is meant by carrier for nutrients?

**Response:** This was indeed a vaguely written sentence. We revised it as "Studies have found that water released by HR can elevate ammonification, N mineralization, and plant inflorescence N uptake (Cardon et al., 2013) and enhance the overall nutrient mobility in dry soil patches (Matimati et al., 2014)."

7. L50: These sound like points that should come earlier in the the introduction and re-introduced in the discussion. They're both really nice points, but perhaps the authors can weave them into the text more smoothly.

**Response:** We moved this up in the introduction. It now appears at the end of the first paragraph and as "In addition to natural systems, such adaptation likely plays a critical role in dry-land farming and rangelands."

8. L56: You should make the first figure that you introduce appear as figure 1.

**Response:** We updated and moved Figure S1 to the main document as Figure 1 (see below). We also added new table (see Table 1) in which treatment descriptions and acronyms are defined and water and nutrient inputs are reported.

9. L64: C1? Why not just C? The different treatments should be stated explicitly before stating them.

**Response:** See above response. There are two different versions of control treatments. Now that these are defined in the text, Figure 1 and Table 1 at the beginning of the methods section it will be easier to follow.

10. L64: Perhaps you could state what the benefit of these redundancies. You have sensors for water content and matric potential. Do you have a SWC for your silica mixture? It could be useful for inferring impacts of rhizosphere features on soil water movement.

    **Response:** The justification for using two different sensors is because the two approaches are well suited to different ranges of soil moisture. Psychrometers provide better sensitivity to small changes in soil moisture when the soil is very dry, with accurate measurement range of $-50\,\mathrm{kPa}$ to $-8000\,\mathrm{kPa}$. Whereas dielectric sensors have poor sensitivity to small changes in water content, particularly when the soil is dry. Therefore, we changed the sentences to

    "Dielectric water content sensors (5TE of Meter, Pullman, WA) were placed at the center of each compartment at $14$ cm below the surface to capture the bulk-scale soil moisture dynamics. At the same soil depth, the dry compartments of treatment **D** and **C1** were outfitted with pairs of thermocouple psychrometric water potential sensors (Psypro of Wescor Inc. Logan, UT) to measure the localized soil water potential with very high degree of sensitivity (Brown, Bartos, 1982; Andraski, Scanlon, 2002; Whalley et al., 2013). The combination of the two sensor types allows quantification of water dynamics with high degree of fidelity from the wet to dry moisture range.

11. L95: Please provide some details regarding what these tests are.

    **Response:** In this study, Welch's ANOVA was used to test whether there are statistically significant differences between the means of plant performance indicators in the three treatment groups (treatment **C1**, **C2**, and **D**). A regular one-way ANOVA assumes homoscedasticity or homogeneous variance inside each test group; however, in our study, indicators showed different variances inside each

treatment. For example, the fruit dry mass of C2 showed much larger variances than the other treatments. The Welch's ANOVA allows the test for groups with heteroscedasticity or heterogeneous variances inside each testing group. The Games-Howell test was used to rank the means of plant performance indicators in three treatment groups. We changed the sentence to

"Plant physiological indicators were compared across treatments using a Welch's analysis of variance (ANOVA) to avoid interference from heteroscedasticity of those indicators (Welch, 1947) and posthoc Games-Howell test for multiple comparisons from R (Games, Howell, 1976)."

12. L101: This line isn't very clear. Could you elaborate in a few more sentences? Also, are you making a distinction between soil physical properties near and away from the rhizosphere? If so, it might be worth stating this explicitly and later highlighting these differences in a table.

**Response:** The "rhizosphere wetting" here was referred to as the "HR magnitude." We carefully checked the manuscript to replace different terms with a consistent term of HR magnitude throughout the manuscript. We performed the SWC measurement of pure sand with nutrient solutions with the same concentrations for the irrigation. According to Reviewer #3, we changed the "rhizosphere water" content to "root zone water content". More detail that was previously provided on the supplemental material is now added to the methods section.

13. L108: Your labeling system has to be clearly defined earlier in the text. It might be worth just making a short table that indicates the different experimental scenarios.

**Response:** We added a Figure 1 and a Table 1 to provide more details. Treatments are described in detail at the beginning of the methods section. See above responses.

14. L111: This statement sounds a bit redundant with the methods and materials

section. Just state what's being plotted in figure 1.
**Response:** Removed as suggested.

15. L114: It might be worth moving Figure S2 back in here. It seems to merit its place in the main text.
**Response:** We modified and moved Figure S2 to the main document as Figure 3.

16. Figure 1: Describe the figure a bit more. Perhaps state why some of these results have massive spreads. Also, what's the difference between the points and the points with the uncertainty ranges? I know you're trying to demonstrate that there are no discernible trends based on these results, but do not be cavalier when presenting this data. Also, if you really don't think it's worth plotting the data in Figure S2, then it might be worth stating it only briefly. Otherwise, I think it would be nice to integrate the information with this figure.
**Response:** We added more detail to the figure caption (now appears as Figure 2)
"Figure 2. Comparison of plant physiological indicators (a) total dry biomass, (b) fruit dry mass, (c) number of flowers, (d) total N uptake, (e) N uptake in Fruits, and (f) N use efficiency in treatment **D**, **C1**, and **C2**. The orange dots represent values of individual replicates. The white diamonds and whiskers represent the mean and standard deviation within each treatment. Distribution of N content along the canopy length is shown in Figure 3. One of the replicates in treatment **C1** did not produce fruits, resulting in larger deviations in fruit dry mass and N update in treatment **C1**. "

17. L118: I would refrain from using language that is so astringent. You are still operating under an artificial set up with a select cultivar. Your data stands strong on its own, so there's no need for the hard sell.
**Response:** Removed "unequivocally" as suggested.

18. L122: This reads as though it belongs in the intro. I would suggest moving it there and rephrasing this intro.
**Response:** We moved and incorporated it into introduction as suggested.

19. L126: Remarkable has a connotation of suspense and surprise. Perhaps rephrase to,
"Results highlighted that the density of roots in the wet and dry compartments are indistinguishable, despite the vast disparity in water availability." Your results speak for themselves. They require no added shock value.
**Response:** Revised as suggested.

20. L127: This belongs in materials and methods.
**Response:** Moved to methodology as suggested.

21. L128: What does this mean? Should there have been a 3D root architecture? The figure illustrates the photo of the set up.
**Response:** Figure 4a shows the bulk distribution of roots in the dry and wet compartments. However, the spatial distribution of the roots prior to excavation is not visible in this photograph because the roots settled at the bottom after the soil was excavated. The paragraph was revised as follows. This photograph was not adding a significant information to the story of the paper, so has now been moved to supplemental material. It is now presented along with other photographs of experiments including the roots of the seedlings before transplantation to the split pots.

22. L131: None of the distributions in the figure look uniform. (f) and (h) even have slight bumps towards the centerline of the set up. I would re-phrase this to more accurately describe the results.
**Response:** The Stacked bar-chart format used previously was hiding the pattern. See revised Figure 4 below. We also add references to the specific part of each figure as we describe the different treatments.

23. L134: As stated before, both seem to exhibit this concentration in the midsection. Perhaps I'm misunderstanding the figure. If so, please try to elaborate in a way that is clear and consistent with the figures.
**Response:** See the above response.

24. L138: This is a generalization for specific soil conditions and nutrients. For instance, phosphorus is highly susceptible to binding on finer soil particles, so you have to be careful with these broad brush strokes and generalizations on the basis of your set up.
**Response:** We agree with the reviewer that chemical retention might have hindered the leaching of strongly sorbing nutrients in natural systems, such as phosphorus, as the reviewer mentioned. The statement is specific to the soil conditions used in this study. We added a sentence to the discussion section:
"The coarse silica sand used in this study likely facilitated leaching of reactive nutrients such as phosphorous."

25. L140: Is this shown in any of the figures or tables? Please cite the figure where this is highlighted.
**Response:** We added cross-reference to Figure 4b.

26. L145: Authors need to reference their figures more thoroughly and provide some explanation for what the reader's should be paying attention to. For instance, root hairs appear to be more prominent in the dry cases in both scenarios. Though I am skeptical as to what mechanisms are at hand (i.e. root hairs lack the vasculature for rapid water movement), they must be doing something in dry soils (perhaps the structures alone are at play). It's worth noting this. Furthermore, the ordering of your figures is chaotic. They're good figures, but the authors need to organize them how they appear.
**Response:** We splitted previous Figure 4 into Figure 4 and Figure 5. We then reorganized Figure 5, so that the parts are ordered in the sequence they are referenced. We added cross-references to each component as they are discussed. All the SEM images have identical magnification that permits visual qualitative comparison. We noticed two patterns. First, there are more root hairs in the dry compartment than the wet compartment of both treatment **D** and **C1**. Second, the nutrient-rich dry compartment appears to have much higher density and thicker root hairs than the nutrient-free dry compartment. Thus, our interpretation of these observation is that increase in the density and thickness of root hairs is a response to availability of nutrients in dry soil.This observation of increased root-hair growth as adaption to nutrient availability has been reported in (Bates, Lynch, 2001; Zhang et al., 2018). These points have been made clear in the revised manuscript.

27. L150: This again reads like a piece from the methods and materials. This should be moved accordingly.
    **Response:** Moved as suggested.

28. L157-159: Please reference a figure that will illustrate this comparison. Also, this doesn't seem to suggest hydraulic redistribution, but it does imply a more sophisticated control over root water uptake. It would seem that the atmosphere doesn't have such a complete tyranny over transpiration. ... I don't understand this argument. Could you please elaborate as to why this should be the case?
    **Response:** This sentence is supported by comparing the water content dynamics in the 'dry' compartments of treatment **D** and **C1**, which are depicted by the red lines in Figure 6a and 6b, respectively. Although the 'dry' compartments of both treatments received equal amounts of water, the rates of drying were vastly different. The water content in the nutrient-rich compartment (Figure 6a) dropped to pre-irrigation level within one day while the water content in the nutrient-poor compartment declined slowly over several days. Thus, water redistribution within the compartment was ruled out as a primary factor. We attributed the faster rate of soil moisture decline in the nutrient-rich compartment to root uptake, which is

also consistent with the higher density of roots in this compartment (Figure 4b). Later in the manuscript, we argue that HR prevented the soil in the nutrient-rich compartment from progressively drying towards permanent wilting point, thereby enabling maintenance of root function in the dry spells between irrigation.

29. L162: Were the SWC's determined on the rhizosphere soil? If so, please state this clearly in the methods.
**Response:** The SWC of the clean sand was determined using dew-point potentiometry (WP4c, Decagon Devices, Pullman, WA). We used nutrient solutions with the same concentrations as the the irrigation water used in the dry compartment of treatment **D**. Because the principles of measurement that apply to WP4c and the *in situ* psychrometers are identical, the resulting SWC can be used to reliably convert water potential to water content. Possible effects of rhizosphere structure development on SWC is not accounted for and this assumption is considered acceptable as the psychrometer placement does not target rhizosphere soil. To clarify our definition as suggested by Reviewr #3, we changed the term "rhizosphere" to "root zone". These statements have been added to the methods section in the main body.

30. L163: So there was little redistribution. Isn't this apparent in Figure 3?
**Response:** Yes, it was apparent in Figure 3 (now Figure 6) that the water content reaches and is maintained at a constant value one day after irrigation. The dielectric sensors are not able to resolve the minute changes in water content that result from HR. The water content dynamics derived from the psychrometer data Figure 7b shows this HR derived dynamics clearly. The point of this statement is that HR does not significantly contribute to the **net transpiration** by the total plant as water released by HR at night is taken up again the next day. We made this point to highlight the fact that function of HR is not necessarily to increase water uptake as has been postulated elsewhere (Ghezzehei, Albalasmeh, 2015; Carminati et al., 2016).

31. L166: However, your results do not clearly point to this. Thus why should this be the conclusion drawn?
    **Response:** Our results show (a) increased root proliferation in dry nutrient-rich compartments, (b) rapid uptake of nutrient-rich irrigation water, (c) HR only in the presence of nutrients in dry compartments, (d) more vigorous root-hair development in nutrient-rich dry compartments, and (e) no significant difference in nutrient uptake whether nutrients are applied in dry or wet soils. When taken together, these observations suggest that HR water is likely playing an important role in supporting the observed root and root-hair growth and in increasing nutrient mobility. The discussion was revised here and elsewhere to make these inferences clearer.

32. L167: You never mentioned root hair proliferation in dry regions (I think I did!). Please do this earlier on and reference the figure. Furthermore, it seemed that it was the case for both D and C1, so does it really matter if it's nutrient rich? Lastly, why should this imply that HR was essential? If anything, these results do not illustrate (at least not clearly) that HR is even happening.
    **Response:** This is now addressed to revisions to the methods section described above.

33. L170: It appears to be showing water fluxes. Make sure this is being cross referenced appropriately. Also, Figure 4 (c) needs a bit more detail. What are the different colors? Please include the text in the figure.
    **Response:** Cross-referencing was corrected. The caption and methods section were revised to explain the HR fluxes."

34. L171: You mean the inward flux? HR has not been defined very clearly.
    **Response:** HR magnitude was defined as the water flux released from the root surface to the soil (mm/day). The ambiguity was clarified in the revised methods section.

35. L174: It doesn't make sense to me. What's the benefit of HR if it's facilitated under wetter conditions? It doesn't even seem like redistribution if it moves water better when the regions are wetter. Perhaps the dynamics play a role in the redistribution? If you have SWC for the rhizosphere soil, then shouldn't you be capturing most of the effects of the exudation, root hairs, and rhizosheath? It is interesting to suggest that the plants are more actively moving water, but I don't think that this evidence is quite strong enough or developed to make those claims. I think the authors should focus on the water uptake that they presented earlier, as those were strong indicators that plants have the capacity to selectively control the how they're taking up water. The HR portion doesn't hold up as strong on the basis of their results.

A similar comment also appears in the conclusion

L216: The logic here isn't tight. Think about the infiltration experiments in unsaturated soil. The onset, water movement is rapid because of capillarity. If you have water being released from roots, that water should be pulled out more rapidly in the onset. It's not an intuitive problem, but the argument doesn't appear sound.

**Response:** The observed pattern can be explained if we assume the water flux at the root-soil interface is governed by a flux law that is analogous to Buckingham-Darcy law

$$q = \frac{\psi_r - \psi}{\delta} K(\psi)$$

The first term denotes the water potential gradient, i.e., difference between the value at the root surface ($\psi_r$) and the rhizosphere ($\psi$). $\delta$ denotes the thickness of the rhizosphere and $K$ is hydraulic conductivity of the rhizosphere. Therefore, the relationship between $\psi$ and HR flux depends on both hydraulic conductivity and hydraulic gradient. Notice that drying of the rhizosphere soil affects these two factors in opposite directions–by increasing the gradient and decreasing the conductivity. Therefore, the net effect should be dependent on the relative magnitudes of these effects. For simplicity, if we assume that $K$ is described by

$K = K_s e^{\alpha(\psi - \psi_o)}$. Then, the above flux law can be simplified as

$$\frac{\delta q}{K_s} = (\psi_r - \psi)e^{\alpha(\psi - \psi_o)}$$

It can be shown that the above scaled flux has a maxima at $\psi = (\alpha\psi_r - 1)/\alpha$. Above this threshold, rhizosphere drying would increase flux while below the threshold the opposite would occur. In the range of measurements observed in this study ($-1000\,\text{kPa} \leq \psi \leq -100\,\text{kPa}$), the latter appears to dominate. Moreover, we want to point out that the analogy with infiltration suggested by the reviewer does not fully apply to the rhizosphere condition. In a typical infiltration experiment, the water potential at the wetting front is at or near zero, whereas during HR, the water potential at the root surface is much lower than zero. Therefore, a more appropriate analog is internal redistribution, which generally slows down with soil drying.

36. L186: This is a strong result.

37. L188: First off, this is a monster sentence. Please break it up. Second, it wasn't clear to me throughout the entirety of this document what pulses were until here, so that needs to appear MUCH earlier. Lastly, make it clear to the reader why this should be the case and why this should be important. Why would plants need to briefly wet regions that have nutrients?
**Response:** We added more details in the introduction to describe the process and impacts of HR in nutrient uptake. In the methodology, we clarified the terms of intermittent irrigation, HR magnitude. We further carefully checked and replaced the terms that can be potentially confusing to readers with clearer and consistent terms.

38. L193: "...within arbitrary discontinuous subregions of the rooting zone." That's a bit wordy too, isn't it? Perhaps find a way to better emphasize that water and

nutrients can be remote and disconnected.
**Response:** We changed to
"within the spatially segregated rooting zone."

39. L194: This is not a paragraph. Link it with the previous one.
**Response:** Revised as suggested.

40. L199: This is a pretty vague statement. Which signaling feedbacks are you describing in particular? After describing them, please explain how you came to those conclusions.
**Response:** We added a potential mechanism and reference after the sentence:
"A recent study reported that spatial availability of water is a key trigger for biosynthesis and transport of root-inductive signal compounds (Bao et al., 2014)."

41. L200: There were equal amounts, but they were spatially distributed very differently. I think the stronger way to form this argument would be to stating again that the plants ability to acquire their nutrients and thrive despite having them segregated and disconnected highlight more complex mechanisms at play during root water/nutrient uptake. Try to emphasize the disconnection.
**Response:** We revised the sentence as suggested. The sentence we added was:
"The marked differences in root mass distribution between the two compartments of the three treatments demonstrated that plants' ability to acquire water and nutrients and thrive, despite having received them from spatially disconnected soil regions. This observation highlighted a complex whole-plant scale regulation of root growth and functions."

42. L202: I'm not sure that you're truly monitoring the signaling with your set up. However, I think it's fair that you report the proliferation that you see and cite studies that link this to signaling.
**Response:** We added a potential mechanism and reference after the sentence:

[Figure]

"This nutrient-driven response is consistent with Weidlich (2018), where roots of non-legume plants were found actively proliferate toward N-producing legume plants."

43. Figure 5: This is very unclear. What do you mean by elevated water retention? Is the soil retaining more water? What is the basis of this? Water release by diurnal cycles makes sense, but use the language correctly. Elevated water retention is out of context here.
**Response:** We changed the "elevated water retention" to "elevated water content".

44. L204: Those aren't your results. Can your results confirm this claim? Otherwise, I suggest leaving it out.
**Response:** We removed the reference and changed the sentence to
"... HR to maintain root function of effective water uptake in dry soil patches..."

45. L205: You never make this link clear in the results section, and I'm not sure if you actually can. You will need to argue for why this is necessarily the case. Maybe consider the kinds of enzymatic processes associated with nutrient uptake by plants. It's likely that these reactions require some degree of moisture in the soil.
**Response:** This was addressed in our previous responses to comments on L40 (introduction) and L166 (results).

46. L208: Make this clear when you present the result
**Response:** We added a sentence in the result section: "The soil water potential in the dry compartment of treatment **D** fluctuated in a diurnal pattern with daytime decrease and nighttime increase. The fluctuation magnitude ranged between $-100$ to $-1000$ kPa, which was above the permanent wilting point $-1500$ kPa. In contrast to that, no nocturnal increase in soil water potential was observed in the dry compartment of treatment **C1**."

47. L211: I would remove this list. It doesn't seem to help the flow.
**Response:** Removed as suggested.

48. L216: The logic here isn't tight. Think about the infiltration experiments in unsaturated soil. The onset, water movement is rapid because of capillarity. If you have water being released from roots, that water should be pulled out more rapidly in the onset. It's not an intuitive problem, but the argument doesn't appear sound.
**Response:** The response to this comment was merged with response to comment on L174.

49. L224: So take the arguments and synthesize them in the text so when the reader gets to this point they will be able to agree with you.
**Response:** We followed the reviewer's suggestion and added one sentence that synthesized the connection between HR and root exudation. The sentence we added was,
"The elevation of water retention in the rhizosphere potentially increased the soil moisture compared to bare soils, which assisted the occurrence of HR through enhancing the soil hydraulic conductance."

50. L226: This is good to point out. Try and hypothesize some of the differences that might occur under less ideal conditions and provide a rebuttal as to why this is a sufficient set up to make the claims that your study is making.
**Response:** Some of the key differences from real conditions include (a) the bulk of plant nutrients are likely to exist in non-available form affixed to mineral surfaces and/or as part of organic matter. Therefore, the role of HR in facilitating nutrient uptake from dry soil patches will depend to the degree at which HR can alter sorption-desorption reactions and rates of mineralization. Therefore, the effective water potential range within which meaningful benefit of HR can be realized is likely to depend on the specific mineralogy, organic matter content and composition, and microbiome. In our experiments, nutrient was delivered with

weekly pulses of irrigation, albeit in small quantities. This was necessary because the medium we chose does not contain any nutrient. In real conditions, such pulses are likely going to be less frequent. Whether HR alone can sustain root functions for extended period of dry spell requires further investigation. These differences and their implications have been added to the paragraph.

51. L233: Remove "multiple lines of"
**Response:** Removed as suggested.

**Figure Captions**

[revised manuscript text omitted]

Distributed
(**D**)

90% water    10% water
0% nutrient  100% nutrient

Control 1
(**C1**)

90% water    10% water
100% nutrient  0% nutrient

Control 2
(**C2**)

50% water    50% water
50% nutrient  50% nutrient

30.5

20.3

**soil volume** = 2 x 2800 cm$^3$

Water content sensor

Psyhcrometer

Datalogger

**Fig. 1.**

[Figure]

[Figure]

**Fig. 2.**

Fig. 3.

[Figure]

Fig. 4.

[Figure]

**Fig. 5.**

[Figure]

**Fig. 6.**

[Figure]

**Fig. 7.**

[Figure]

(a)

[Figure]

Fig. 8.

[Figure]

---

## Author Comment (AC3) · 21 Jul 2020

This manuscript presents experimental evidence that plants can satisfy their water and nutrient demand from mismatchingly distributed water and nutrient resources, if the overall available amount is sufficient. The plant adaptation strategies and regulating mechanisms related to this are discussed. Overall, this is a well-designed contribution of high interest. However, the methods in part lack clarity, and the results and discussion are in parts too speculative.

My first two points are about nomenclature:

**Comment 1**: The first is the definition of the term rhizosphere. There are different ways in literature how to use the term rhizosphere and thus I think it is important to

define clearly what this term means in this paper. I think this paper rather means a part of soil which has a high root density, i.e. it is more used in the meaning of "root zone". That could be confusing as a lot of other work understands the rhizosphere much more locally in form of gradients in the concentration of root-influenced solutes or other compounds extending from the root surface to the 'bulk' soil (Darrah et al., EJSS 57, 2006).

**Response**:We thanked the reviewer's suggestions. We clarified the definition of the term "rhizosphere" in the methodology section. See more details in our reply to Comment 3.

**Comment 2**: The second is the term "exudates". It is often used quite differently in different papers. I rather tend to distinguish "root exudates" as low-molecular weight organic carbon (such as citrate, sugars) and mucilage. An overarching term that includes both exudates and mucilage would be "rhizodeposition" (Oburger and Jones, Rhizosphere 6, 2018). I encourage the authors to also use this nomenclature.

**Response**: We thanked the reviewer for providing suggestions and relevant references. According to the suggestion, we carefully checked the terms throughout the manuscript and replaced "root exudation or mucilage" with "rhizodeposition." The reference was added to the introduction when rhizodeposition was first mentioned in the manuscript.

Some methodological aspects were also not clear to me:

**Comment 3**: I could not find in which depths the water potential sensors were installed. I could also not infer in how far it is justified to call the resulting value a "rhizosphere" water potential. Is it not rather the water potential in the soil layer that has the highest root length density? One could understand this from your sentence on page 3, line 64:

": : :to measure the water potential of the root zone". Comparing the water content that was computed from the rhizosphere water potential (Fig. S5c) with the water content that was measured with the dielectic water content sensor that was installed in the middle of the compartment (Fig. S3a), I can hardly see a difference.

**Response**: We changed the term "rhizosphere" to "root zone" in the result section. We discussed our results that inferred rhizosphere activities in the discusion section. In terms of sensor locations, both psychrometric and dielectric sensors were installed at the same depth of $14$ cm from the soil surface. We added more details to the methodology session about how and where the potential water sensors were installed accordingly. A new figure that illustrates experimental design was added for this revision (See Figure 1 below).

**Comment 4**: How can you be sure that the water increase in the root zone with highest root length density results from HR? Root water uptake and injections will create water potential gradients within one compartment that could result in redistribution of water in the soil.

**Response**: In principle, internal redistribution from moist soil to dry rhizosphere can result in a signature that looks like the trend we observed. However, we ruled out this process for several main reasons. First, the psychrometers were installed prior to root growth around them. If the source and sink for the redistribution were within the mid-section of the dry compartment, then we would have expected to see an out of phase fluctuation in at least one sensor. All the five sensors that functioned well during the experiment showed consistent nightly increase and daily decrease in water potential. Therefore, internal lateral redistribution could not have been the cause of the observed pattern. Second, vertical redistribution (that flow

of water from above or below the densely rooted zone) is possible but not likely. In the treatment **C1**, root density was low in the mid-section. The water added at a weekly cycle was likely being redistributed up and down by capillarity and gravity. But the rate of this transfer is very slow as evidenced by the water content sensors, despite the water content being at a higher level than was observed in the dry compartment of treatment **D**. Therefore, we would expect vertical redistribution to be rather slow process and cannot explain the water potential fluctuation in the dry nutrient-rich compartment. Third, if you zoom in the dielectric water content sensor data, there appears a trace of fluctuation that is consistent with the water potential data. The dielectric sensors were installed vertically have effective volume of measurement that extends beyond the densely rooted zone. Therefore, if there was redistribution from above or below, these trends would not visible. That said, the level of fluctuations we observed is at the detection limit of the sensors. Finally, the water potential fluctuation pattern that we observed is consistent with our previous field observations (different plant, but similar textured soil and using the same sensors). In the previous study, we definitively concluded that HR was occurring using isotope tracing (Bogie et al., 2018). We supplied deuterium labeled water directly to deep roots from sealed vials, with no path other than the root for uptake. We detected the label in neighboring shallow-rooted plants within hours for several days, clearly showing that HR occurs in sufficient quantity to be able to be taken up by shallow rooted plants. When taken together, these arguments strongly support what we observed was indeed HR.

**Comment 5**: The structure of the paper needs attention. I suggest, for example, to move the paragraph lines 105-115 page 4 to the description of the split-root experiment in the Methods section. Then, the "D" and "C1" will be easier to understand in line 64 on page 3.

**Response**: We moved the paragraphs, as the reviewer suggested. In addition, the criticism of manuscript structure and organization was well addressed according to all three reviewers. More details can also be found in our general replies to Reviewer #1 and #2.

Some claimed results seem a bit too speculative to me:

**Comment 6**: Comment 6: The reason for root accumulation at the bottom could also just be that the pot was too short. I.e., if almost all the carbon in C1 is invested in the wet and nutrient rich compartment, it may be possible that the roots would have grown much deeper than in the other treatments if they had been given the space.

**Response**: The reviewer was correct; we agreed that if the chamber had been open, the roots would grow deeper in the wet compartment of treatment C1. Due to the same reason, roots could grow deeper in both compartments of treatment C2. However, rather than understanding the constraints on rooting depth from the physical barrier, we tried to focus on how nutrient and water distribution drive the root distribution. The roots accumulated at the bottom of a close-end chamber, or alternatively, roots grew deeper in an open-end chamber; both would suggest that, presumably, roots extract the water and nutrients leached to the deeper soil layers given higher soil moisture conditions.

**Comment 7**: Comment 6: "Moreover, multi-scale signalling and feedbacks appear to be involved": How could you support this statement with your results?

**Response**: We added a potential mechanism and reference after the sentence:
"A recent study reported that spatial availability of water is a key trigger for biosynthesis and transport of root-inductive signal compounds (Bao et al., 2014)."

**Comment 8**: While it is known that hormonal signaling may regulate the transpiration demand at the leaves, the water flow into our out (HR) of the roots follows (passively) local hydraulic gradients between xylem and soil (e.g. Rothfuss and Javaux, Biogeosciences 14, 2017). What regulation mechanisms exactly do you mean by your statement "HR is biologically-mediated"? Would that be regulation of root hydraulic properties? How could you support that with your results?

> **Response**: We agreed with the reviewer that HR magnitude is biophysically regulated by the water potential gradients and conductance of plant-soil systems. However, our results suggested that the occurrence of HR correlated with nutrient availability because HR was observed only in the dry nutrient-rich patches but not nutrient-deficient ones. The nutrient enrichment drove the root growth, which builds a conductive bridge between the wet and dry compartments and eventually allows the occurrence of HR. Therefore, there is a plant-scale decision making, or biologically-driven decision-making that regulates the occurrence of HR, driven by the plant nutrient demands.
>
> We expanded the discussion to clarify and emphasize the importance of biological regulation of HR occurrence by adding a sentence:
> "As opposed to nutrient-deficient dry soil patch, the apparent occurrence of HR in nutrient-rich dry soil patches was probably a consequence of plant nutrient demands and extensive root distribution."

Responses to minor comments are provided blow each comment. To facilitate review of our responses, we added all the figures at the end of this document. We added three figures during this revision. Most figures have been revised and the captions have been expanded and clarified.

**Minor comments**

1. I could not see that the number of replicates was mentioned in the Methods section.
   **Response**: For each treatment, there were three replicates. We changed the sentence at L56 to
   "Our experiments were conducted using laterally split soil compartments arranged in three treatments with three replicates of each treatment, as depicted in Figure 1."

2. P3 L57: How long did it take the plants to reach that height?
   **Response**: We changed the sentence to
   "Tomato seedlings were germinated in potting mix and grown for $3$ weeks until they reached $5 - 10\ cm$ in height."

3. P3 L59: How many roots were there at this stage? Was the tap root recognizable and was there a strategy to place it into a specific compartment?
   **Response**: We provided a photo of the seedlings the root mass at the time of transplantation. The is added to the supplemental material as Figure S7. We did not differentiate the taproots from the other roots for the split-root experiments. We divided and the roots without consideration of which side was going to be in specific compartment. Any possible influence of initial root mass differences to be small and randomly distributed between treatments and compartments. The description now includes clarifications. To assist the review of our response, we show Figure S7 as Figure 9 in the response (see below). In the manuscript, Figure S7 will be presented in the supplemntal materaials.

4. P3 L85: When you scooped out the soil, did you cut the roots within these 2cm intervals?
   **Response**: Yes, we cut the roots within each interval to obtain the root mass

distribution in each soil layer. To clarify that, we changed the sentence to
"The coarse root pieces in each interval were cut and removed ..."

5. Fig. 2a: I do not see the relevance of Fig. 2a. I also suggest to split the rhizoshelth and root mass distribution to two separate figures. Fig. 2h: dry and wet labels are confusing for this treatment.
**Response**: We moved previous Figure 2a to the supplemental materials and split subfigures of root mass distribution and SEM images into two single figures.

6. P8 L165: "taken up by the roots".
**Response**: Changed as suggested.

7. P11 L211: The absence of HR in C1 was not mentioned in the Results section.
**Response**: We added a sentence to the result section at L161 as
"The soil water potential in the dry compartment of treatment **D** fluctuated in a diurnal pattern with daytime decrease and nighttime increase. The fluctuation magnitude ranged between $-100$ to $-1000\ kPa$, which was above the permanent wilting point $-1500\ kPa$. In contrast to that, no nocturnal increase in soil water potential was observed in the dry compartment of treatment **C1**."

**Figure Captions**

1. Schematic illustration of the experimental design. Each pot consists of two isolated compartments that are fused together by glue. Roots of seedlings were roughly divided half and a half during transplantation. The experiment consisted of one treatment in which the bulk quantity of water and nutrients were distributed separately (treatment **D**) and two control treatments in which nutrients were applied with most of the water. In Control 1 (**C1**) water was applied non-uniformly as in **D**, whereas in Control 2 (**C2**), water and nutrients were applied uniformly to

both compartments. Placement of sensors and water and nutrient delivery tubes are illustrated. The diagram is not to scale.

[revised manuscript text omitted]

**Distributed (D)**
90% water 10% water
0% nutrient 100% nutrient

**Control 1 (C1)**
90% water 10% water
100% nutrient 0% nutrient

**Control 2 (C2)**
50% water 50% water
50% nutrient 50% nutrient

30.5

20.3

**soil volume** = 2 x 2800 cm$^3$

Water content sensor

Psyhcrometer

Datalogger

**Fig. 1.**

**Fig. 2.**

Fig. 3.

[Figure]

Fig. 4.

100 µm

**(a) (b)**
**(c) (d)**

**Fig. 5.**

[Figure]

**Fig. 6.**

[Figure]

**Fig. 7.**

[Figure]

**(a)**

Water

Nutrient
Solution

HR

**Day**

nutrient
uptake

**Night**

HR

Elevated
water
content

**(b)** Adaptation to mismatched
resource distribution

Transpiration

Nutrient
uptake

HR

Nutrient-rich
dry layer

Nutrient-poor
wet layer

**(c)** Distributed water and
nutrient delivery design

Water

Nutrients

**Fig. 8.**

[Figure]

**Fig. 9.**

---

## Author Comment (AC4) · 23 Jul 2020

**Interactive comment on: "Root uptake under mismatched distributions of water and nutrients in the root zone" by Jing Yan et al.**

Response to Reviewer Comments #1

July 23, 2020

**General Comment**

The manuscript entitled "Root uptake under mismatched distributions of water and nutrients in the root zone" aims to test how mismatched distribution of water and nutrient influence nitrogen acquisition and plant growth. The authors further investigate how hydraulic redistribution and changes in root morphology can explain their results. While the objective of the study is very relevant and rather clearly defined and justified, the rest of the manuscript (material and methods, results, discussion and conclusion) is hard to follow, with crucial elements lacking from the material and methods. It makes it difficult to understand why the authors did some measurements and what they really measured. In the discussion and conclusion, I found some part too speculative. For example, how could you conclude so strongly about the crucial role of root hairs and production of mucilage, only based on non-quantitative microscopic observations? I suggest you better describe what you really demonstrated and what your results only

suggest. Overall, I think that the data provided here are of good quality, that the design was well though, but the manuscript is poorly written. See specific comments to help you to improve it.

**Response**: The criticism of this reviewer was shared by the other two reviewers as well. The manuscript has been revised with this in mind. Details that were previously included in the supplemental materials are now added to the materials and methods section. As Figure 1, we added a schematic diagram that describes the treatments and placement of sensors. Details were added to captions, and the results are explained thoroughly. Finally, we revised the results and discussion sections to avoid over-interpretation. Specific suggestions and comments from all reviewers were helpful in making these edits.

Specifically, in the conclusion section, we synthesize the observations from study and offer a conceptual model of what we believe is a depiction of nutrient and water dynamics in natural environments where nutrients and water may exist in a distributed fashion. Some of these statements are hypotheses and require further testing. We make it clear when we have direct evidence, and when we are speculating. For example, the role of HR in nutrient cycling was not observed in this study. In fact, we intentionally avoided conditions that would complicate the interpretation of where the plants acquired nutrients from. However, it is very likely that HR plays a crucial role in mineralization when nutrients are locked in organic matter in the dry region. This hypothesis is, in part, supported by previous studies that documented HR in arid regions, where the soils are coarse-grained and of low nutrient content. The synthesis of our knowledge is presented in Figure 8a and 8b. We believe this synthesis is an important contribution that can serve as a launching point for further studies.

Responses to specific comments are provided below each comment. To facilitate the review of our responses, we added all the figures at the end of this document. We added three figures during this revision. Most figures have been revised, and the captions have been expanded and clarified.

**1  Specific Comments**

Abstract

Please precise which plant (or at least type of plant) you grew as I am not sure that trees, herbs and grass shows the same adaptations to mismatches. At least, it should be proven before concluding it. We lack the experimental design (at least briefly mentioned) in the abstract

> **Response**: We revised the abstract by adding the plant species, i.e., tomato plants, and a brief description of experimental design.

**L.13 – 15** : It is too strong from my perspective. You did not quantified root hair density, neither production of root mucilage.

> **Response**: Yes, we agree with the reviewer that SEM and confocal microscopic images did not provide quantitative information on root hair density and mucilage content. However, we believe the indirect and qualitative evidence gleaned from these observations is essential in deciphering how the plant root functions under mismatched conditions. Reviewer #2 commented on the role of root hairs at L145 and suggested presenting and discussing the observation of root hair enrichment in dry compartments of treatment **C1** and **D**. Therefore, we removed the description of root hair and mucilage

from the abstract and conclusion while emphasizing this topic's discussion. At L220, we revised the discussion:

"..two possible pathways might have allowed roots to modify rhizosphere hydraulic properties...". Besides, we revised our description of the results of root hair. In brief, we reported that root hair density appears to be denser in the dry compartments of treatment **D** and **C1** and root hairs appear to be thicker in the nutrient-rich dry compartment (treatment **D**), compared to the nutrient-poor dry compartment (treatment **C1**).

Introduction

**L. 28 – 31** : I was pleased to read that you mention the role of rhizospheric soil microbes to make nutrients available for plants. This could, and for my perspective should, be mentioned in the discussion too (although not too extensively as you did not measured any microbial parameter here). You mention specific adaptations of plants to water or nutrient deficiency (or heterogeneous distribution), namely: (i) Preferential growth in moist areas and modifications of root exudation (l.32-36) and hydraulic redistribution (l. 38 – 42). In these two paragraphs, you develop more adaptations to water scarcity or heterogeneity in fact. Adaptations to N deficiency or heterogeneity are less developed. For example, roots of a non-legume plant can forage toward the roots of a legume plant (Weidlich et al., 2018). Associations with soil microbes, such as N-fixing bacteria and mycorrhizae are as well strategies to enhance N acquisition and avoid growth limitations. Differences in root morphology (SRL, ratio root length/dry mass) of absorptive roots are typically used to describe foraging behavior of roots to acquire root N (a mobile nutrient). Proliferation of root hairs (which is not mentioned here, although it seems to be important for your article), or root clusters (highly branched roots) are more known to enhance acquisition of P, a less mobile element often found in patches (Lambers et al., 2011; Bates et al., 2001). With regards to adaptations of roots to water scarcity, see as well the recent article from Bristiel et al., (2019). The adaptations cited

here do not sufficiently cover the topic.

1. Weidlich, E. W., Temperton, V. M., & Faget, M. (2018). Neighbourhood stories: role of neighbour identity, spatial location and order of arrival in legume and non-legume initial interactions. Plant and Soil, 424(1-2), 171-182.

2. Lambers, H., Finnegan, P. M., Laliberté, E., Pearse, S. J., Ryan, M. H., Shane, M.W., & Veneklaas, E. J. (2011). Phosphorus nutrition of Proteaceae in severely phosphorus impoverished soils: are there lessons to be learned for future crops?. Plant Physiology, 156(3), 1058-1066.

3. Bates, T. R., & Lynch, J. P. (2001). Root hairs confer a competitive advantage under low phosphorus availability. Plant and Soil, 236(2), 243-250.

4. Bristiel, P., Roumet, C., Violle, C., & Volaire, F. (2019). Coping with drought: root trait variability within the perennial grass Dactylis glomerata captures a trade-off between dehydration avoidance and dehydration tolerance. Plant and soil, 434(1-2), 327-342.

**Response**: Thank you for the references and additional adaptation mechanisms. We have provided a more extensive introduction and discussion regarding the role of root morphology, microbial activities, root nutrients in nutrient foraging.

In the introduction (L37), we added:
"Strategies of root foraging toward local soil nutrient deficiency or heterogeneity can be more divergent. Such strategies can involve the proliferation of root branches, root hairs. For example, the occurrence of hairy roots and root clusters has been reported enhancing phosphorus acquisition (Lambers et al., 2011; Bates and Lynch, 2001). Furthermore, the association with N-fixing bacteria and mycorrhizae has been found essential

in root growth and N acquisition. The root interaction between neighboring plants further complicated our understanding. For example, a recent study showed that the roots of a non-legume plant forge toward the neighboring legume plant roots, where nitrogen is locally enriched (Weidlich et al., 2018)."

**L. 50 – 53:** While the objective was rather clearly described, I do not see the point with these last sentences.

**Response**: We moved this up in the introduction. It now appears at the end of the first paragraph and as
"In addition to natural systems, such adaptation likely plays a critical role in dry-land farming and rangelands. "

Material and methods

In general, this section lack clarity and there is several important missing information.The methods are often described without explaining their aim. The subsection 2.1 (which could be renamed experimental design) lack to present the experimental design. Instead, the signification of treatment D, C1 and C2 is given at the beginning of the results! I can't find figure S1. I lack as well the number of replicates. The duration of the experiment should be given here too. The quantities of N, water, how are loss compensated, where it is added should be described: : : please report what was done with accuracy.

**Response**: We added a more detailed description of the methodology section. In addition, we moved Figure S1 from the supplemental materials to the main document and added a table that summarized water and nutrient application for each treatment.

Revision included:

"Our experiments were conducted using laterally split soil compartments arranged in three treatments with three replicates of each treatment, as depicted in Figure 1. ... The experiment lasted for $140$ days with a total application of the $653 - 676 \ mm$ water and $120 \ mg$ N. The compartment-specified application schemes were reported in Table 1."

**L.62 – 67**: the measurement of water content and water potential belong to plant and soil characterization

**Response**: The revision of the materials and methods section has addressed this.

**L. 80**: Please define NUE, I guess this is nitrogen use efficiency, but this should be written.

**Response**: We now provide the definition of acronyms when they are first introduced.

**L. 86**: What do you mean by "further gravimetric measurements"?

**Response**: We clarified the procedure by changing it into "gravimetric quantification of root mass".

**L.88-93**: It is not clear why you are doing these microscopic analyses. Why laser of two different wavelengths are used? What is gold coating for?

**Response**: The electron and confocal microscopic analysis provided complementary evidence about the morphological adaptions of roots and rhizosphere. Specifically, while SEM images provided detailed surface information with a higher spatial resolution, a confocal microscope differentiates

the autofluorescent root compounds and non-fluorescent soil matrix. Gold coating (sputtering) is a standard technique in SEM imaging, especially when samples are non-conductive and sensitive to beam damage, including most biological samples. A conductive homogeneous layer of gold provides sharp images while maintaining the integrity of the sample morphology. We added references in the main document that provide these justifications (Kim et al., 2010; Golding et al., 2016). The confocal microscope shoots the incident light with shorter-wavelength ($405\ nm$ in this study) to excite the fluorescent emission from the plant root tissues and other organic compounds roots released. It then captures the emitted signals with longer-wavelength (488 nm in this study). We used this technique to distinguish autofluorescent compounds (roots and other organic compounds) from the non-fluorescent sand matrix. We added more details in the methodology to justify using these imaging techniques:

"$405\ nm$ and $488\ nm$ lasers were used to excite and acquire autofluorescent compounds from the roots that distinguish from the non-fluorescent soil matrix. ... We then used SEM imaging to gain detailed surface information of the rhizosheath with a higher spatial resolution. ... A homogenized gold coating was used to provide a conductive layer of metal that enhances image quality by preventing charging and damage (Kim et al., 2010; Golding et al., 2016)."

Results

The subsections are confusing. Is plant water and nutrient uptake (3.3) not related to plant physiology (3.1)? Please reorganize. Moreover, some parts belong to material and methods, other to discussion. Focus on what you have observed here.

**L.105 — 115**: This belongs to material and methods.

**Response**: Moved to the methodology section as suggested.

**L. 118 – 120**: This is your interpretation of the results. It should go to discussion.

**Response**: Moved to the discussion as suggested.

**L.122 – 124**: This belongs to introduction

**Response**: Moved to the introduction as suggested.

**L. 127**: How did you test that root density do not differ between the two compartments? By comparing root masses? If this is the case, it is thus not root density but root mass. Moreover, in table A2, the wet and dry compartments of the treatment D are significantly different.

**Response**: We agreed with the reviewer that statistically more root mass was found in the dry than wet compartment in treatment **D**. We changed the sentences to:
"Results highlighted that $60\%$ of cumulative root mass grow into the dry compartment of the treatment **D**, despite the vast disparity in water availability."

**L. 127 – 128**: this belongs to material and methods

**Response**: Moved to the methodology section as suggested.

**L. 130-131**: Belongs to material and methods

**Response**: Moved to the methodology section as suggested.

**L. 131 – 134**: Please indicate what this higher root masses in the deeper part suggests in the discussion. Here you should describe the results.

> **Response**: We moved the
> "... suggesting slight ..." to the discussion.

**L.135**: Again root density or root mass?

> **Response**: We changed it to
> "root mass".

**L.136- 138**: again, belong to discussion. Moreover, avoid detailing twice the same idea. An increase in root mass in the deeper layer is seen in the three treatments D, C1 and C2.

> **Response**: We moved it to the dicussion and carefully removed the redundancy.

**L. 138 – 140**: This should be stated in material and method, not here.

> **Response**: Moved to the methodology section as suggested.

**L. 140**: Did you measure root growth? Or are you indicating root mass? Root mass is not equal to root growth as the root mass at a given point depends on root growth, and root death (life span / root turnover).

> **Response**: We changed the "root growth" to "net root mass increase".

**L.143 – 145** from "which is: : :" belongs to discussion.

**Response**: We moved it to the dicussion as suggested.

**L.146**: How did you measure root hair density? What test did you do to conclude for significant differences?

**Response**: We did not measure the root hair density quantitatively; instead, the results were based on the visual comparison. To avoid further confusion, we remove the word "significantly", which implies qualitative comparison.

**L. 147 – 148**: This belongs to discussion

**Response**: Moved to discussion as suggested.

**L.150**: Avoid starting a new paragraph with "the above observations". It suggest you are still developing previous ideas, so why starting a new subsection?

**Response**: Revised as suggested.

**L. 151 – 153**: Belongs to introduction

**Response**: Moved to introduction as suggested.

**L.155**: Did you describe the frequency of the irrigation?

**Response**: After the plants become established over the first two weeks, the application of water or nutrient solutions in wet compartments of treatment **D** and **C1** were provided daily, while in the dry compartments, a small volume of water and nutrient solutions were provided once a week. For treatment **C2**, water was applied daily, while nutrient solutions were provided once a week. The total amounts of water and nutrient application are now presented in a new Table 1. Moreover, Figure S1 was revised added to the main body of the manuscript as Figure 1 and shows the irrigation pattern. We added descriptions of the frequency of irrigation and nutrient application events in the methods section.

**L. 157**: The information about the frequency of N addition should be given in material and method.

**Response**: See above response.

**L. 161**: How did you converted soil water potential data to rhizosphere water content?

**Response**: The soil water retention curve was determined independently using the same sand used in the experiment. We used dew-point potentiometry (WP4c, Decagon Devices, Pullman, WA) and nutrient solution identical to the irrigation water used in the dry-compartment of treatment **D**. Because the principles of measurement of WP4s and psychrometer are identical, we were able to convert the results of soil water potential measured from the psychrometers to soil water content by using the soil water retention curve. The description of the method, the fitting of the data and it use are now explained in the Methods section.

**L. 163 – 164**: Avoid opinion terms such as "closer inspection".

[Figure]

**Response**: Edited a suggested.

**L. 166**: Do no cite literature in the results, you should describe what you found here.

**Response**: Edited as suggested.

**L. 168**: What do you mean by "habitable environment"? For the roots? For rhizospheric microbes? Your focus here is not nutrient uptake, please stay stick to it.

**Response**: Although our main story is about nutrients, the questions we aim to address include understanding the mechanisms by which nutrient uptake from dry soils is possible. This includes understanding how roots are able to survive and grow in dry soils to the extent that was observed in this study. Our observations suggest that HR prevents the soil from progressively drying towards a stage that could hamper root function. In Figure 7a, notice that because of the contribution of HR, the water potential fluctuated between $-800$ kPa and $-600$ kPa, but did not dry further than that. Moreover, maintaining the soil water status above a detrimental threshold would permit soil microbes to carry out essential nutrient cycling functions in the rhizosphere. Therefore, we believe the function of HR in nutrient uptake is closely tied to the contribution of HR to the habitability of the rhizosphere to roots and microbes. We have made these linkages between HR, rhizosphere habitability, and nutrient uptake more clear and coherent in the revised manuscript.

**L. 173 – 175**: Again, this is not the description of the results.

**Response**: We removed the sentence as suggested.

**L. 176 – 179**: This belongs to discussion

**Response**: We moved the sentence to the dicussion as suggested.

**L. 180**: Do you assume that the organic coating is root mucilage? How did you quantified it? What are the two fluorescent wavelength for?

**Response**: The microscopic image provides only qualitative information about the changes in root and rhizosphere morphology. The organic coating provided evidence that the modification of root hairs and rhizodeposition on rhizosphere soil properties, which has been reported in previous studies (Koebernick et al., 2017, 2019; Carminati et al., 2010; Ghezzehei and Albalasmeh, 2015). The fluorescent wavelength distinguishes the fluorescent root compounds, including both root tissue or amorphous rhizodeposition from the non-fluorescent soil matrix. More details can be found in our reply to L88-93.

**L. 181**: keep suggestion to the discussion

**Response**: We moved the sentence to the discussion as suggested.

**L. 181**: This is an interpretation, not a result.

**Response**: We moved the sentence to the discussion as suggested.

Discussion

**L. 184–186**: This belongs to introduction

**Response**: We moved the sentence to the introduction as suggested.

**L.192**: How could you confidently conclude that plant performance are less sensitive to localized scarcity in water and N if nutrient and water are sufficient in other locations where the roots forage. You did not tested it. To know it you should have a mismatched distribution of water and nutrients, with an overall limitation in water and N (compared to your treatment D).

**Response**: We clarified this statement. The experimental designed included compared localized nutrient deficiency in wet environment (treatment **D**) and localized water deficiency in nutrient poor environment (treatment **C1**) with non-deficient uniform resource availability (treatment **C2**). Our observations (see Figure 2 and 3 in the revised version) show no significant differences between these three treatments. This lead to our conclusion that under experimental conditions we tested localized deficiency of nutrients and water did not have measurable impact of overall plant performance. We think that the above ground performance (greenness, biomass, flowering, fruits, nutrient content) were indistinguishable despite the considerable differences in resource distributions is an important finding that is supported by multiple measurements. We did not intend to address overall nutrient and/water limitations in this study. All plants received equal amounts of water and nutrients. The added elaborations and reorganization of the sequence of presentation will make the objective and experimental design clearer and consistent with the conclusions that we came up with.

**L. 194**: I can't see what allow you to draw this conclusion here. Nothing written in the paragraph above allow to conclude it, although I think that you are right to point different plant strategies in case of mismatches.

**Response**: This conclusion was based on observed differences in root distribution between compartments and within compartment; qualitative observation of root-hairs; and presence of HR. Specifically, HR was induced and

dense growth of thick root-hairs was observed only when the plants had to rely on nutrients that were concentrated in dry soil. This strategy absent in plants grown without such spatial of nutrients from water (in the same soil and under the same total nutrient and water availability).

**L. 197–198** : Sentence not clear

**Response**: The paragraph is now rewritten with more elaboration.

**L. 198** : You did not measure root proliferation as far as I have understood and what do you mean by this term: root growth? Root turnover?

**Response**: We changed "root proliferation" to "the extensive root mass distribution in the dry nutrient-rich soil compartment..."

**L.199**: What is multi-scale signaling and feedback? This is too vague.

**Response**: We added more elaboration and listed potential mechanisms. A recent study reported that the spatial availability of water is a key trigger for biosynthesis and transport of root-inductive signal compounds (Bao et al., 2014).

**L. 200**: You did not describe root allocation in the results. You surely want to say that this is the relative mass of roots in the two compartments? Or in the various depths? Please specify it. I can't see how it points a whole plant scale regulation of root growth. Please explain.

**Response**: We revised the paragraph to emphasize the differences in root mass distributions between the treatments. The revised results section
and clearly show the differences between the final root mass distributions. We use consistent terminology throughout the revised manuscript and use cross-references to direct the reader to the data that is the basis of the discussions and conclusions.

**L. 201**: This confirms the foraging behavior of non legume roots to legume roots (Weidlich et al. 2018).

**Response**: We agree.

**L. 204–205**: This is one of the most interesting result of the study. Please detail more.

**Response**: We expanded our discussion, as suggested. We now include a discussion on how the root mass and root-hair density observed under the dry nutrient-rich environment were much higher than the nutrient-free environment subjected to similar water application regimen. Moreover, we reiterated that HR was observed only in the former case. These observations were the basis for concluding that HR plays an important role in supporting the growth and maintenance of roots in an otherwise non-conductive dry environment. It is also important to note that because of HR, the water potential did not progressively decline in the intervening period between the weekly irrigation with nutrient solution.

**L. 206**: What do you mean by "vigorous"? How did you measure it? It is not clear to me how drying after wetting event can indicate vigor.

**Response**: We replaced 'vigorous' with 'more effective in water uptake'.

**L. 214**: This is an important result too.

**Response**: We agree. This portion was further clarified and elaborated in response to comments from reviewer #2.

**L. 218–219**: What do the references refer to? You conclude here from your own results and cite the related figure. I guess the references indicate that this has been previously shown?

**Response**: We thanked the reviewer's critical review. Our results were consistent with previous evidence. Therefore, we added an additional discussion to distinguish them from our findings:
"This result was consistent with previous studies, where the loss of hydraulic conductance of soil-plant systems has been attributed to the decline in HR magnitude (Scholz et al., 2008; Ryel et al., 2002; Meinzer et al., 2004)."

**L. 221**: Need a reference

**Response**: We added a reference as suggested (Koebernick et al., 2017, 2019).

**L. 226–229**: Avoid finishing with limitations. Specify them either in the conclusion or in the discussion but not at the end as this is the last take home message for the reader.

**Response**: We moved the discussion of limitations and combined with our suggestion of future study to L192. Please also see our reply to L192.

Conclusion

**L.231**: "could" or "did"? Be clear with what you have demonstrated. In general, better differentiate what you showed and what your results suggests.
**Response**: To clearly present our results,
we changed "plants" to "tomato plants" and "could" to "can."

**L. 243**: How did you measure root activity?

**Response**: The changes in soil moisture reflected the root activity in terms
of water and the associated nutrient uptake. We revised the sentence to
emphasize the root function in water uptake that guides the audience at
L206. Please see more details in our reply at L206.

**L. 244**: What is a vigorous nutrient cycling. Did you measure it?

**Response**: We removed the ambiguous word. In this synthesis part of
the conclusion we are combining what we observed with that is previously
known to suggest possible functions of HR beyond what we observed here.

**L. 250-260**: I enjoyed the final thought about application, but it makes the conclusion
quite long and bring new ideas. This paragraph may be moved to the discussion.

**Response**: We moved the material that describes the potential application
in agricultural context to the end of the discussion section. The revised
conclusion now includes only one sentence that summarizes the agricul-
tural implication.

**Table A1**: I would enjoy a graph or table with the values measured here. N uptake is
central in your article (according to the objectives).

**Response**: We added the mean and standard deviation values of each
variable and treatment reported in Table 2. See at the bottom of this docu-
ment.

**Figure 3**: What does the different color means? It would be better to rename treatments with an easy understandable name, instead of D, C1 and C2, which looks more a code for labeling pots.

> **Response**: This Figure now appears as Figure 6. The treatments and abbreviations have been defined at the outset in the methods section and in Figure 1. The different shades of red and blue in these figures are used to distinguish between replicates. The revised caption addresses these differences.

**Figure Captions**

[revised manuscript text omitted]

Distributed
(**D**)

90% water   10% water
0% nutrient 100% nutrient

[Figure]

Control 1
(**C1**)

90% water   10% water
100% nutrient 0% nutrient

[Figure]

Control 2
(**C2**)

50% water   50% water
50% nutrient 50% nutrient

30.5

20.3

**soil volume** = 2 x 2800 cm$^3$

[Figure]
 Water content sensor

Psyhcrometer

Datalogger

**Fig. 1.**

**Fig. 2.**

[Figure]

Fig. 3.

[Figure]

**Fig. 4.**

(a) (b)
(c) (d)

100 μm

**Fig. 5.**

[Figure]

**Fig. 6.**

[Figure]

**Fig. 7.**

[Figure]

Fig. 8.

---

## Author Response (AR1)

Dear Editor,

Thank you for considering our manuscript for publishing in Biogeosciences. We are writing to inform you that we have submitted our revised manuscript for further review.

The manuscript was thoroughly revised following the suggestions of the reviewers. Significant changes during the revision are summarized below.

a. The structure has a much-improved flow. This includes reorganizing and revising the content throughout the manuscript for consistency and cohesiveness.

b. We give a detailed description of the methods, and the results are fully described. Essential procedure and results are summarized in conceptual schematics, tables in the main document, and appendix.

c. All the figures are reformatted for consistency, and the key findings have been moved to the main body. Captions are updated with sufficient details.

Other specific comments were provided in our replies to each reviewer. Please contact us for additional information. Thank you for your time.

We look forward to hearing from you.

Sincerely,

Jing Yan

Teamrat Ghezzehei

Nathaniel Bogie

[revised manuscript text omitted]

---

## Author Response (AR2)

Dear Editor,

Thank you for considering our manuscript for publishing in Biogeosciences. We are writing to inform you that we have submitted our revised manuscript for further review.

The manuscript was thoroughly revised following the suggestions from the most recent round of review. Significant changes during the revision are summarized below.

a. The responses to comment 4 and 6 of reviewer 3 were now added and incorporated into the discussion (see L269-279 and L265-268, respectively).

b. The response to comment 35 of reviewer 2 was added to Appendix A.

b. We changed the legend of Figure 4 and 6 in treatment **C2** from "wet" and "dry" to "left" and "right" to avoid further confusion. Accordingly, we thoroughly reviewed the manuscript for consistency of term usage, including changing the labels in Table 1, legends in Figure S6, and captions in Figures 4 and 6.

Please contact us for additional information. Thank you for your time.

We look forward to hearing from you.

Sincerely,

Jing Yan

Nathaniel Bogie

Teamrat Ghezzehei

[revised manuscript text omitted]